# Operando neutron diffraction reveals mechanisms for controlled strain evolution in 3D printing

A. Plotkowski [1] ✉, K. Saleeby[2], C. M. Fancher [1], J. Haley [3], G. Madireddy[2], K. An [4], R. Kannan[2], T. Feldhausen [2], Y. Lee[5], D. Yu[4], C. Leach[3], J. Vaughan [2] & S. S. Babu[2,6]

Residual stresses affect the performance and reliability of most manufactured goods and are prevalent in casting, welding, and additive manufacturing (AM, 3D printing). Residual stresses are associated with plastic strain gradients accrued due to transient thermal stress. Complex thermal conditions in AM produce similarly complex residual stress patterns. However, measuring real-time effects of processing on stress evolution is not possible with conventional techniques. Here we use operando neutron diffraction to characterize transient phase transformations and lattice strain evolution during AM of a low-temperature transformation steel. Combining diffraction, infrared and simulation data reveals that elastic and plastic strain distributions are controlled by motion of the face-centered cubic and body-centered cubic phase boundary. Our results provide a new pathway to design residual stress states and property distributions within additively manufactured components. These findings will enable control of residual stress distributions for advantages such as improved fatigue life or resistance to stress-corrosion cracking.

Residual stresses are ubiquitous in modern manufacturing processes[1,2]. In many cases, they are mitigated through design or post-treatment processes which are usually either applied to the entire material uniformly (e.g., annealing) or limited to near-surface regions (e.g., shot peening)[3]. Alternatively, residual stress distributions could be controlled throughout the material to locally induce tensile and compressive stresses for macro-scale performance gains[4,5]. Dictating the final residual stress distribution in a component, however, requires detailed control of thermal distributions during processing or the development of materials whose physical properties and phase transformations modify the stress state in desirable ways. A key example of the latter approach is the development of low-temperature transformation (LTT) steels for fillet welds. In this case, volume

expansion during the austenite-to-martensite transformation leads to compressive stresses in the weld[6,7]. For idealized uniaxial constraint during cooling (e.g., a Satoh test[8]) LTT steels exhibit an increase in compressive stresses during the martensite transformation and a residual stress state that is increasingly compressive as the transformation temperature is depressed.

The emergence of additive manufacturing (AM), in which components are sequentially fabricated in a layer-by-layer fashion, has led to renewed focus on management of residual stress[9,10]. Complex thermal cycles in AM often produce nonequilibrium microstructures and properties that differ from conventional manufacturing routes[11], requiring significant effort to achieve homogeneous properties[12]. However, the site-specific nature of material deposition and heat input

[1]Materials Science and Technology Division, Oak Ridge National Laboratory, Oak Ridge, TN 37831, USA. [2]Manufacturing Science Division, Oak Ridge National Laboratory, Oak Ridge, TN 37831, USA. [3]Electrification and Energy Infrastructure Division, Oak Ridge National Laboratory, Oak Ridge, TN 37831, USA. [4]Neutron Scattering Division, Oak Ridge National Laboratory, Oak Ridge, TN 37831, USA. [5]Computational Sciences and Engineering Division, Oak Ridge National Laboratory, Oak Ridge, TN 37831, USA. [6]Department of Mechanical, Aerospace, and Biomedical Engineering, The University of Tennessee – Knoxville, Knoxville, TN 37996, USA. ✉e-mail: plotkowskiaj@ornl.gov

also offers opportunities to manipulate local conditions to modify microstructure and properties[13–17]. This approach may be viewed as a form of material-structure-performance integrated manufacturing, characterized by a desire to obtain "the right materials printed in the right positions"[18]. In principle, this approach may include the design of residual stress distributions, for example to tune tensile-compressive asymmetry[9] or prevent fatigue crack initiation and slow fatigue crack propagation[19]. Recent research has investigated the use of LTT steels in AM processes to achieve unique and complex residual stress states[20–22]. Yet, process conditions[22] and phase transformations[23] have complex interacting effects on the resulting residual stress distribution, which are further dependent upon thermal contraction, accumulation and annealing of plastic strain, and the temperature dependence of the stress state and material properties. Unlike the Satoh test, the thermal stresses in AM are transient and multiaxial, thermal gradients are high, and multiple rapid heating and cooling cycles are often experienced. To demonstrate a pathway to controlling the residual stress distribution under these complex conditions, it is necessary to characterize the temperature, phase, and stress state during processing.

## Results

### Measuring and controlling residual stress in AM

The residual stress distribution within an AM component is a manifestation of gradients in accumulated plastic strains when those strains are incompatible with any possible elastic displacement field[1]. Therefore, direct observation of the strain evolution during AM would have a profound impact on the ability to tune processing conditions and materials design to achieve residual stress distributions tailored for a given application. The total strain as a function of time, $\varepsilon_{tot}(t)$, is described here as the summation of thermal ($\varepsilon_{th}$), elastic ($\varepsilon_{el}$), and plastic ($\varepsilon_{pl}$) contributions:

$$\varepsilon_{tot}(t) = \varepsilon_{th} + \varepsilon_{el} + \varepsilon_{pl}. \tag{1}$$

Note that volume change associated with phase transformations is implicitly included in Eq. 1 in its observable effect on the surrounding elastic and plastic strains in the constituent phases. Computational modeling may separate these contributions but also requires detailed experimental observations for validation purposes. Experimental techniques such as center hole drilling[24], digital image correlation[25] or fiber optic strain sensors[21] are either applicable only for post-process characterization or are not easily decomposed into components of the strain response. X-ray characterization has been used for operando interrogation of rapid physical processes[26–28], but a high-energy source is required for diffraction studies of bulk complex geometries. These setups introduce significant measurement complexity, e.g., transmission geometry, that prohibits the measurement of lattice strains for specific locations within the structures of interest. By comparison, neutron diffraction (ND) is well suited for operando measurements and allows for location specific characterization. ND has been previously used for ex situ characterization of AM parts[29,30], in situ study of friction stir welding[31], and some initial operando AM characterization[32]. Here, we demonstrate use of a custom AM system for operando ND characterization, which, when coupled with thermal imaging and computational modeling, yields insight into strain component evolution during processing of an LTT steel.

### Operando neutron measurements and analysis

A wire-arc AM system was used to fabricate mild and LTT steel planar walls in three sections consisting of seven layers each (Fig. 1A). Between the build sections, active cooling was applied by forcing compressed air through the aluminum substrate to cool the samples to room temperature. This build design was used to intentionally produce cyclic heating and cooling to elucidate the effect of processing on

residual stress evolution. Ex situ neutron diffraction maps (measured at the high intensity diffractometer for residual stress analysis (HIDRA) at the High-Flux Isotope Reactor (HFIR) at Oak Ridge National Laboratory[33,34], Fig. 1B) were used to characterize the residual elastic lattice strains along the centerline of each alloy. In both materials, the lattice strain varies spatially over the height of the sample. However, the amplitude of these cyclic variations is larger and more pronounced in the LTT steel and contrary to the predominantly tensile residual strain in the mild steel, alternates between tensile and compressive strains. Since both mild and LTT steels undergo cyclic phase transformations between FCC and BCC but show different residual elastic strain distributions, it is necessary to understand the pathway to these final states during processing.

Operando experiments were performed to collect neutron diffraction and infrared thermal imaging data during AM (Fig. 1C) at the VULCAN beam line at Oak Ridge National Laboratory's Spallation Neutron Source (SNS). Three in-plane detector banks simultaneously probed three scattering vectors ($\hat{Q}_1$, $\hat{Q}_2$, and $\hat{Q}_3$) to measure corresponding lattice strain orientations using a 5 mm cubic voxel. Figure 1D shows dilatometry data illustrating the phase transformations for the LTT alloy[35], with a martensite start ($M_s$) temperature of approximately 240 °C on cooling, and a temperature at which martensite converts to FCC austenite ($A_s$) on heating at approximately 640 °C. The dilatometry data show a higher coefficient of thermal expansion (CTE) for austenite ($22.33 \times 10^{-6}$ °C$^{-1}$, shown by the slope of the curve on cooling) compared with martensite ($12.42 \times 10^{-6}$ °C$^{-1}$). The room-temperature retained austenite fraction was measured to be ~3 vol% using ND data from a well-annealed sample. On heating, $A_s$ should be considered an "austenite growth" temperature, above which significant changes in phase fraction are expected, but without the necessary requirement for FCC nucleation[36]. The data in Fig. 1D were measured for an as-fabricated WAAM LTT sample and may therefore be considered representative of the transformations observed during fabrication. See Supplementary Note 1 for additional details.

The processing sequence (Fig. 1E) included time-resolved operando measurements during building of each section. Following deposition of each seven-layer section, active cooling was applied to bring the sample to room temperature for lattice strain mapping. Diffraction data collected during processing contain information related to each of the strain components in Eq. 1. Plastic strain affects peak breadth, while elastic and thermal strain contributions cause peak shift. Consequently, plastic deformation may be analyzed separately, but thermal and elastic strains are challenging to distinguish. In the case of an isothermal sample (i.e., after cooling to room temperature), the thermal strain may be neglected, and elastic lattice strains calculated as:

$$\varepsilon_{el}^{L,T} = \frac{d^{\hat{Q}_i}}{d_0^{\hat{Q}_i}} - 1 \tag{2}$$

where $d^{\hat{Q}_i}$ is the interatomic spacing of a given $hkl$ reflection for the longitudinal or transverse direction along $\hat{Q}_1$ and $\hat{Q}_2$, respectively, and $d_0^{\hat{Q}_i}$ is the room temperature interatomic spacing for a stress-free reference. However, to understand the pathway to the final elastic strain distributions, trends in the time-dependent elastic strain must be calculated. For this purpose, the peak shift in detector 3 was assumed to be sensitive only to temperature, that is, the normal elastic lattice strain in detector 3 may be assumed to be negligible. This assumption takes advantage of the transverse strain being small compared to the longitudinal strain, as well as in the direction of the Poisson strain from the stress in the longitudinal direction. As the Poisson strain is opposite in sign from the longitudinal direction, there is a direction, $\theta$, from the longitudinal direction for which the

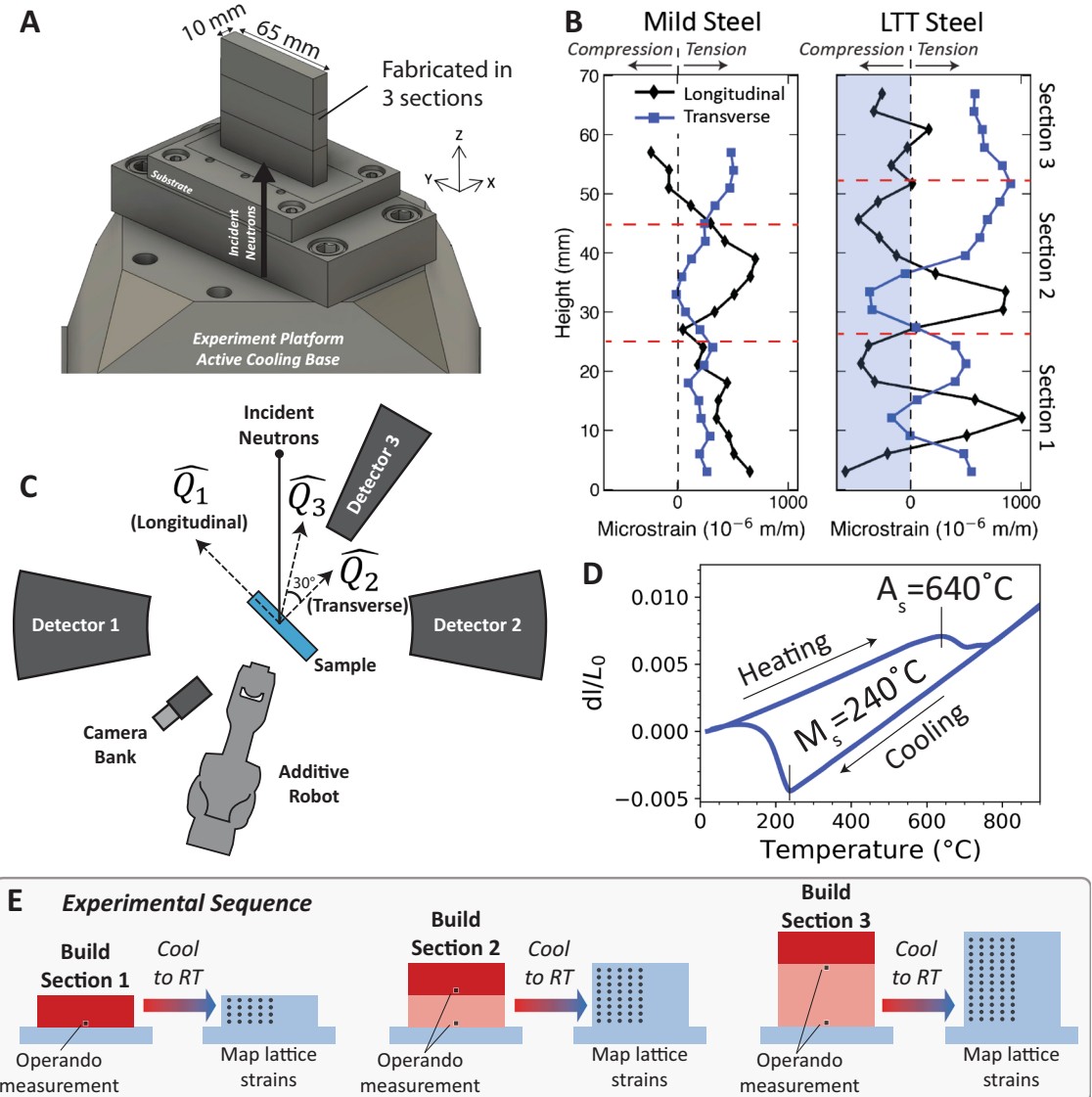

**Fig. 1 | Experimental setup and ex situ data. A** Wire-arc additive manufacturing was performed to produce mild-steel and low-temperature transformation (LTT250) steel samples. Samples were fabricated in three sections and cooled to room temperature between. **B** Ex situ neutron diffraction measurements (collected at HFIR) show elastic lattice strain variation with height along the centerline. **C** An operando additive manufacturing setup was constructed to monitor temperature, phase transformations, and lattice strain during processing at SNS. **D** Dilatometry data showing the approximate phase transformation temperatures for AM fabricated LTT steel. **E** A schematic of the operando experimental procedure which includes builds in three sections, with cooling to room temperature in between using active cooling, following which maps of lattice strain are performed.

strain is zero:

$$\theta = \tan^{-1}\left(\frac{1}{\sqrt{\nu}}\right) \tag{3}$$

where $\nu$ is Poison's ratio. For $\nu = 0.3$, the zero-strain orientation is $\theta = 61.3°$, which is very close to the 60° position of $\hat{Q}_3$ with respect to $\hat{Q}_1$. Vertical stresses may be reasonably neglected as the Poisson effect on the in-plane stresses may be assumed to be isotropic based on the weak texture of the as-fabricated grain structure (see supplementary for details). In this case where the peak shift along $\hat{Q}_3$ may be used as a reference for thermal strain, the longitudinal, $\varepsilon_{el}^L$, and transverse, $\varepsilon_{el}^T$, elastic lattice strains may be calculated using the peak shifts in detectors 1 and 2, respectively, relative to detector 3:

$$\varepsilon_{el}^{L,T} = \frac{d^{\hat{Q}_i}(t) - d^{\hat{Q}_3}(t) + \Delta d_0}{d_0^{\hat{Q}_3}}, \tag{4}$$

where $d^{\hat{Q}_i}(t)$ is the time-dependent d-spacing measured for a given *hkl* plane along the scattering vector $\hat{Q}_i$, where $\hat{Q}_1$ gives the longitudinal strain and $\hat{Q}_2$ gives the transverse strain, and $\Delta d_0$ is a calibration offset (see supplementary for more information). Strains in FCC were calculated using the 311 reflection, and for BCC, the 220 reflection was used.

Equation (3) is quantitatively accurate when $d^{\hat{Q}_3}$ is only a function of temperature. Error in this assumption may be quantified by first considering a temperature estimate for the diffraction voxel if the elastic strain measured by detector 3 is assumed to be small. The time-dependent diffraction peak position, $d^{\hat{Q}_3}(t)$ was converted to temperature using a room-temperature reference d-spacing ($d_0^{\hat{Q}_3}$) and the CTE of the material:

$$T(t) = \frac{\left(\frac{d^{\hat{Q}_3}(t)}{d_0^{\hat{Q}_3}} - 1\right)}{CTE} + 25. \tag{5}$$

Elastic lattice strains along $\hat{Q}_3$ manifest as error in the temperature calculated by Eq. (5). This error propagates to Eq. (4) to affect the magnitude of the elastic lattice strain. Stress in the vertical (build) direction could be a minor cause of such strains, but this stress is otherwise not considered as its effect is to superimpose a small Poisson strain that is measured equally by the three detectors. An error estimate for the strain was therefore made by considering the discrepancy between the ND temperature estimate made with data from detector 3 (using Eq. (5)) and the IR data, and converting the temperature difference to an effective lattice strain to establish quantitative error bands. Note that this approach to error estimation also accounts for inaccuracies in the mean d-spacing in the gauge volume caused by large thermal gradients, as these same gradients will increase the temperature difference between the surface patch measured by IR and the internal temperature calculated from the ND data.

### Time-resolved operando measurements

Time-resolved operando data measured during AM of the LTT steel are summarized in Fig. 2. Calibrated IR data are shown immediately following the end of deposition for each of the three sections. In identical builds, time-resolved neutron data were measured for the selected locations (labeled points 1, 2, and 3) for 5 mm cubic voxels located at the center of the wall during deposition (Fig. 2G shows a schematic of the build dimensions and measurement locations). The time-dependent ND signals are given in Fig. 2 as contour plots of normalized intensity units as a function of d-spacing and deposition time within each section of the LTT steel wall.

Quantitative analysis of the diffraction data required an initial assessment of the crystal structures. The austenite was treated as FCC with corresponding peaks labeled in Fig. 2D. Given the low carbon content in this alloy (0.05 wt%), the c/a ratio for the usual body-centered tetragonal martensitic structure was close to unity (calculated as 1.0016 based on composition[37], and measured as 1.005 based on Rietveld analysis[38,39] of ND data from an annealed sample). Therefore, the low-carbon martensitic structure in the LTT steel was treated as BCC for the remaining analysis. Associated BCC peaks are labeled in Fig. 2D.

At point 1 during deposition of section 1, the material solidifies as FCC (Fig. 2D). The FCC structure is retained during fabrication of the entire section, only transforming to BCC during cooling following the

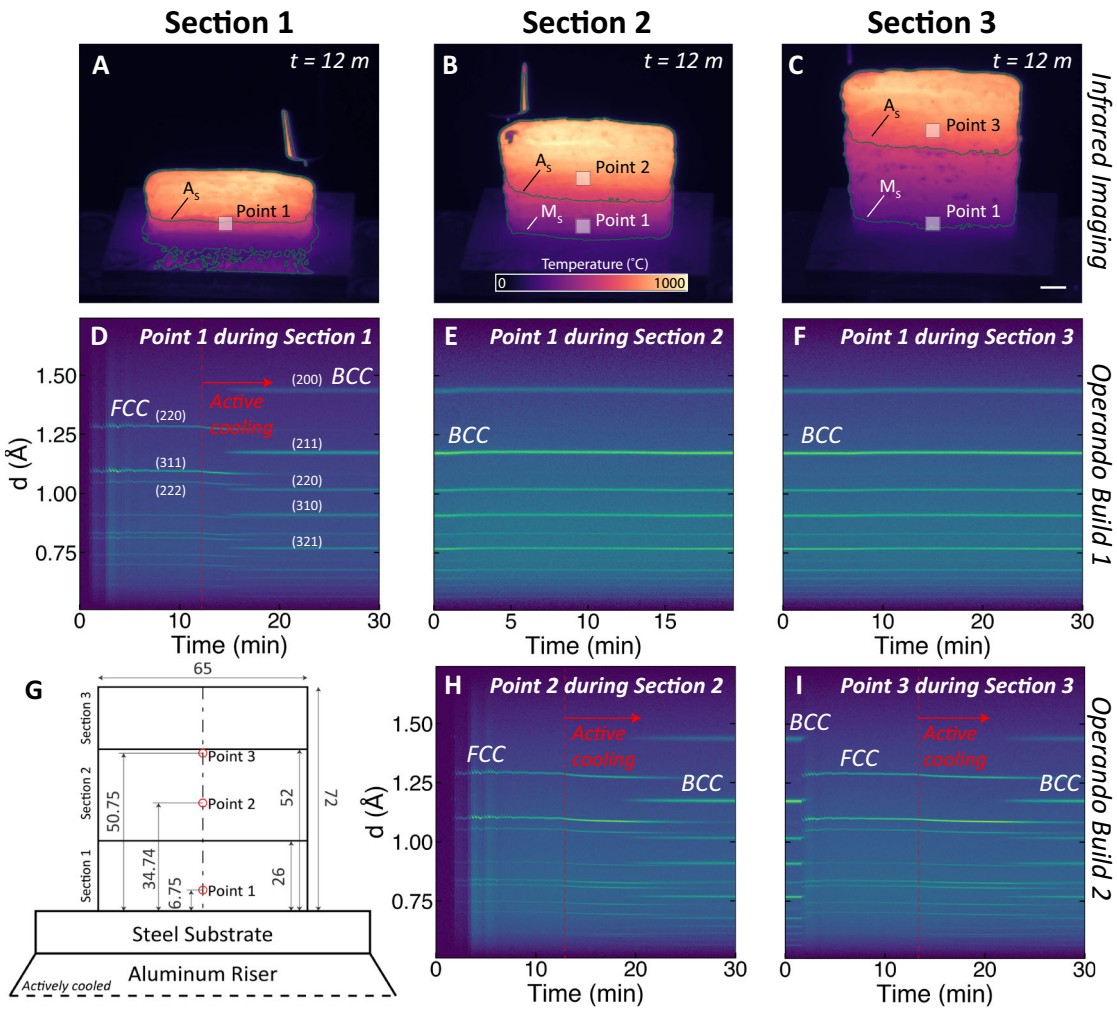

**Fig. 2 | Time-resolved operando measurements.** The temperature distribution as a function of time during manufacturing of the LTT steel was measured using infrared (IR) thermography during **A** section 1, **B** section 2, and **C** section 3 of the build, between which the sample was cooled to room temperature (scale is 1 cm). Time-resolved neutron diffraction data were collected at the indicated points during fabrication of associated build sections. **D** Neutron diffraction data at point 1 during section 1 show peaks for FCC during deposition, and formation of BCC during cooling following completion of section 1. At the same point during fabrication of **E** section 2 and **F** section 3, the temperature does not rise sufficiently to convert the BCC back to FCC. **G** A schematic of the build geometry shows the location of the neutron data collection relative to the build sections; and for **H** point 2 during section 2 and **I** point 3 during section 3, similar trends are observed to those at point 1, in which the material initially solidifies as FCC and transforms to BCC following the onset of cooling. All diffraction data are shown with arbitrary normalized intensity units.

completed deposition of section 1. During fabrication of sections 2 and 3 (Fig. 2E, F), point 1 was not reheated to a sufficiently high temperature to trigger phase transformation of BCC to FCC austenite, because the $A_s$ was much higher (640 °C) than the $M_s$ (240 °C). A similar trend is observed for point 2 during deposition of section 2 and for point 3 during section 3. Point 3, however, was also located just below the interface between sections 2 and 3 and was initially BCC at the start of section 3. When the material above was deposited, this point rapidly reheated above $A_s$, transformed from BCC to FCC, and then remained as FCC for the rest of the deposition process.

## Thermal cycles

The thermal cycles observed by the IR measurements are shown in Fig. 3 and compared directly to the temperature calculated from the d-spacing change observed in detector 3. IR data were extracted periodically between each layer deposition and during cooling after each section for surface patches at the same height as the ND voxels (Fig. 3A). Equation (5) was applied separately for 220 BCC and 311 FCC peaks using appropriate reference values to calculate temperature from the ND data. The FCC temperature calculated by Eq. (5) was used to align the IR and ND data in time by minimizing the difference in temperature during the cooling process following the building of each section. Details of this analysis are reported in the supplementary materials. Figure 3B through D show a comparison of the aligned temperature estimate from ND data with the calibrated IR thermal signal for the three measurement points during deposition and cooling of each respective section. Because the pixel size for the IR data was smaller (about 0.55 mm square) than that for the diffracted voxel, the reported temperature is shown as a mean ± a standard deviation at each IR time step based on the collection of IR pixels. Agreement during deposition was only moderate, and also corresponds with a large deviation in the IR data, suggesting that at high temperature surface condition changes violate the graybody assumption used, or that potentially a significant thermal gradient between the surface and the ND gauge volume exists. However, the estimated temperature from the ND shows excellent agreement with the IR data during the cooling process at the end of each section, consistent with a flattening of thermal gradients within the gauge volume and between the center and surface of the sample. In addition, because the analysis was

performed across multiple experiments, these results demonstrate repeatability of the deposition process, which is a necessary condition for our analysis drawn from multiple ND data sets.

The IR data also illustrate trends in thermal cycles that inform our analysis of the phase transformation and lattice strain behavior. Figure 3A shows an example temperature snapshot following deposition of the third section of the build. Figure 3E through G then show the temperature as a function of height and time along the centerline of the wall, including the $A_s$ and $M_s$ temperature isotherms. During section 1, the entire wall is above the $M_s$ temperature when the last layer is deposited; and it cools almost uniformly, transforming from FCC to BCC in a short period of time. During the second section, a portion of the bottom of the wall (including point 1) does not reach the $A_s$ temperature, but a significant fraction does, creating an "austenite reversion zone." The FCC to BCC transformation happens primarily upon activation of active cooling; but unlike in section 1, cooling is noticeably slower at locations further from the cooling effect of the substrate, and the transformation occurs from bottom to top. A similar pattern is observed in section 3, where a second austenite reversion zone is formed, extending down into section 2, and cooling occurs again from bottom and top, but with an overall slower average cooling rate.

## Elastic lattice strain measured by neutron diffraction

The combination of ND and IR data may be used to interpret the distribution and trends in elastic lattice strain throughout the process. Figure 4A shows the elastic lattice strain distributions measured at room temperature following cooling of each of the three build sections. Both longitudinal and transverse strain directions are shown based on scattering vectors $\hat{Q}_1$ and $\hat{Q}_2$, respectively, calculated using the 211 BCC reflection. The 211 reflection was used, as this lattice plane is representative of the bulk residual strain[40]. After deposition of the first section, the elastic lattice strain maps were primarily compressive in the longitudinal direction, and weakly tensile in the transverse direction. Following deposition of the second section, a band of highly tensile elastic lattice strain appears in the longitudinal direction, with compressive strains directly above. Another similar band is observed in section 2 after fabrication of section 3. The time-resolved elastic lattice strains and associated error estimates (given by the colored

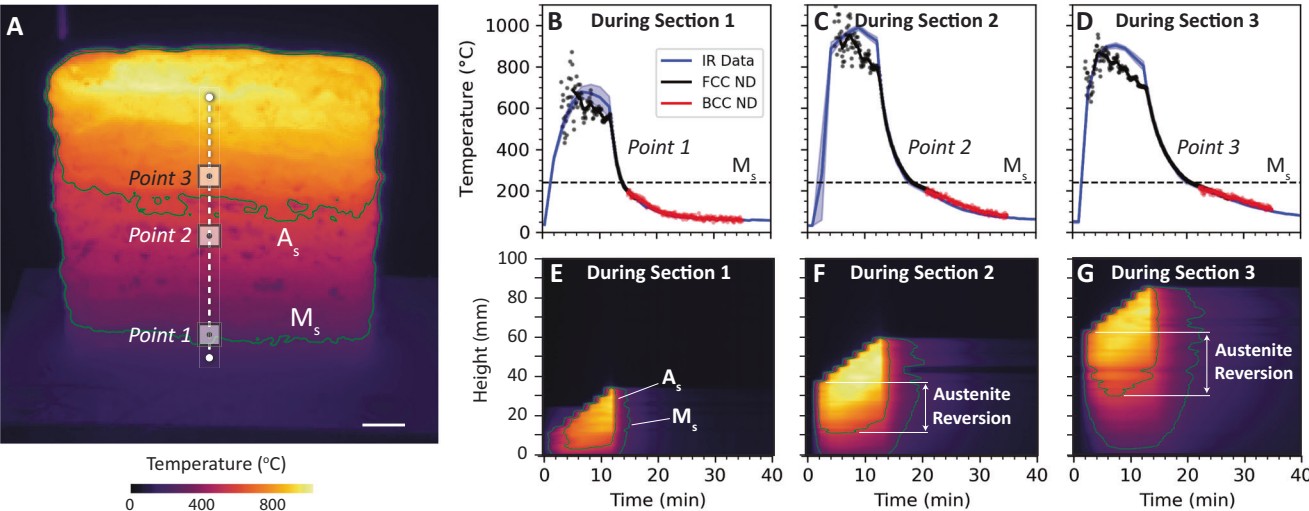

**Fig. 3 | Thermal history during processing. A** Infrared (IR) data were collected during manufacturing between each layer deposition and were calibrated to clearly indicate isotherms associated with the martensite and austenite start temperatures (scale is 1 cm). Temperatures were extracted from the IR data for surface patches associated with the 5 mm voxels for which the time-dependent neutron data were collected. The average for the extracted interpass IR temperature as a function of time (shaded areas show ± one standard deviation based on the IR pixels within the 5 mm surface patch for each IR time step) for **B** point 1 during section 1, **C** point 2 during section 2, and **D** point 3 during section 3 alignment compared well with the ND-calculated temperature histories. The centerline IR temperature along the sample height (indicated by the dotted white line) is plotted for **E** Section 1, **F** 2, and **G** 3, showing regions in which BCC reverts to austenite upon reheating.

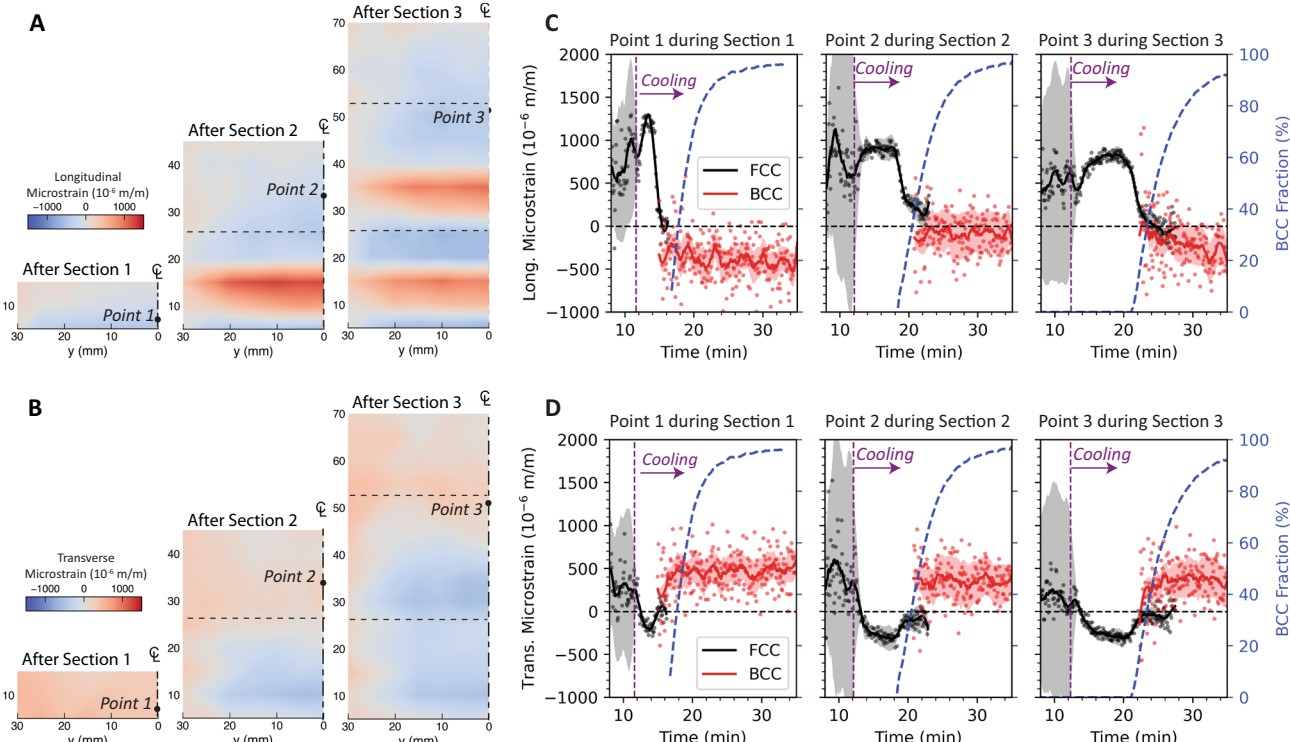

**Fig. 4 | Neutron diffraction lattice strain measurements.** Maps of the **A** longitudinal and **B** transverse residual elastic lattice strain calculated from the 211 BCC peak shift measured with a 5 mm grid spacing after deposition of each section and cooling to room temperature. Time-resolved **C** longitudinal and **D** transverse elastic lattice strains during cooling after each section are calculated based on temperature-corrected reference values using the aligned IR data. Individual data points show the raw data, while lines show data smoothed with a Savitzky-Golay filter[47]. Colored bands indicate errors calculated based on the discrepancy between temperature calculated between detector 3 and the IR data.

bands) are shown during cooling of each section in Fig. 4C, D. The errors in the strain measurements are large during deposition but become small during the cooling process following each section. This result shows that this approach is reliable for measuring the trends in elastic lattice strain contributions during cooling. The time-resolved elastic strain evolution shows a recurring trend. In the initial stages of cooling, the longitudinal strain becomes tensile in all cases. Owing to the Poisson effect, the transverse strains simultaneously become compressive. This effect is consistent with elastic strains caused by thermal gradients and differential contraction even within a single-phase material but is expected to be modified by the presence of both FCC and BCC. With increasing BCC fraction below $M_s$, the longitudinal tensile elastic lattice strain reverses and becomes increasingly compressive. The local strain was thus affected by the phase transformation in two ways: (1) the CTE difference between FCC austenite and BCC martensite reinforces the effects of differential contraction on cooling and (2) the volume expansion from FCC to BCC. These effects tend to oppose one another, with differential contraction causing tensile stress in the FCC during cooling before reaching $M_s$, and expansion during the martensite transformation causing comparatively compressive stress.

Spatial trends are also observed in Fig. 4. The maximum tensile strain in the longitudinal direction during cooling is highest for point 1 and decreases for points 2 and 3. The time between the onset of cooling and the formation of compressive strains is also longer for points 2 and 3 than for point 1. These differences originate from variations in the thermal gradient along the build direction, which are higher for point 1, which is closer to the actively cooled substrate. The high thermal gradient equates to a higher gradient in the martensite fraction, leading to greater tensile stresses in this intermediate stage of cooling. The differences in cooling rate also affect the rate of compressive strain evolution, which is notably slower for

the points further from the cooling and mechanical constraint effects of the substrate.

The transient lattice strain was also calculated for BCC at point 1 during deposition of points 2 and 3 (see Supplementary Fig. 5). The longitudinal strain became tensile during fabrication of section 2 because of the contraction of the austenite reversion zone located directly above it. However, this location returned to a compressive strain in the longitudinal direction after the sample reached room temperature. During fabrication of section 3, the phase transformations and high thermal gradients were far enough from point 1 that the lattice strain was not strongly affected by the processing.

## Phase transformation effect on plastic strain distribution

In addition to the thermal and elastic strain components evident in the diffraction peak shift, the diffraction data combined with thermal cycling provided information on plastic strain evolution. Plastic strain was correlated with the full-width half-maximum (FWHM) of the 311 FCC and 220 BCC lattice reflections. Analysis of the time-resolved data for point 1 during section 1 (Fig. 5C) shows rapid fluctuations in the FCC FWHM during deposition caused by significant thermal gradients present in the gauge volume, followed by an initial reduction during cooling. The FWHM then increases as BCC forms from FCC, consistent with previous work that has shown dislocations form during the transformation from FCC to BCC[41]. This effect is correlated with the volume change from austenite to martensite[42,43].

Deposition of the second section causes a slow decrease in FWHM with time for point 1 (Fig. 5D). This trend can be related to the thermal history using the IR data for points with high (Fig. 5E) and low (Fig. 5F) FWHM following cooling of each section (Fig. 5A). For locations with a large FWHM, final cooling starts from FCC and transforms to BCC, with an associated accumulation of dislocations. By comparison, regions with low FWHM are reheated during deposition of subsequent

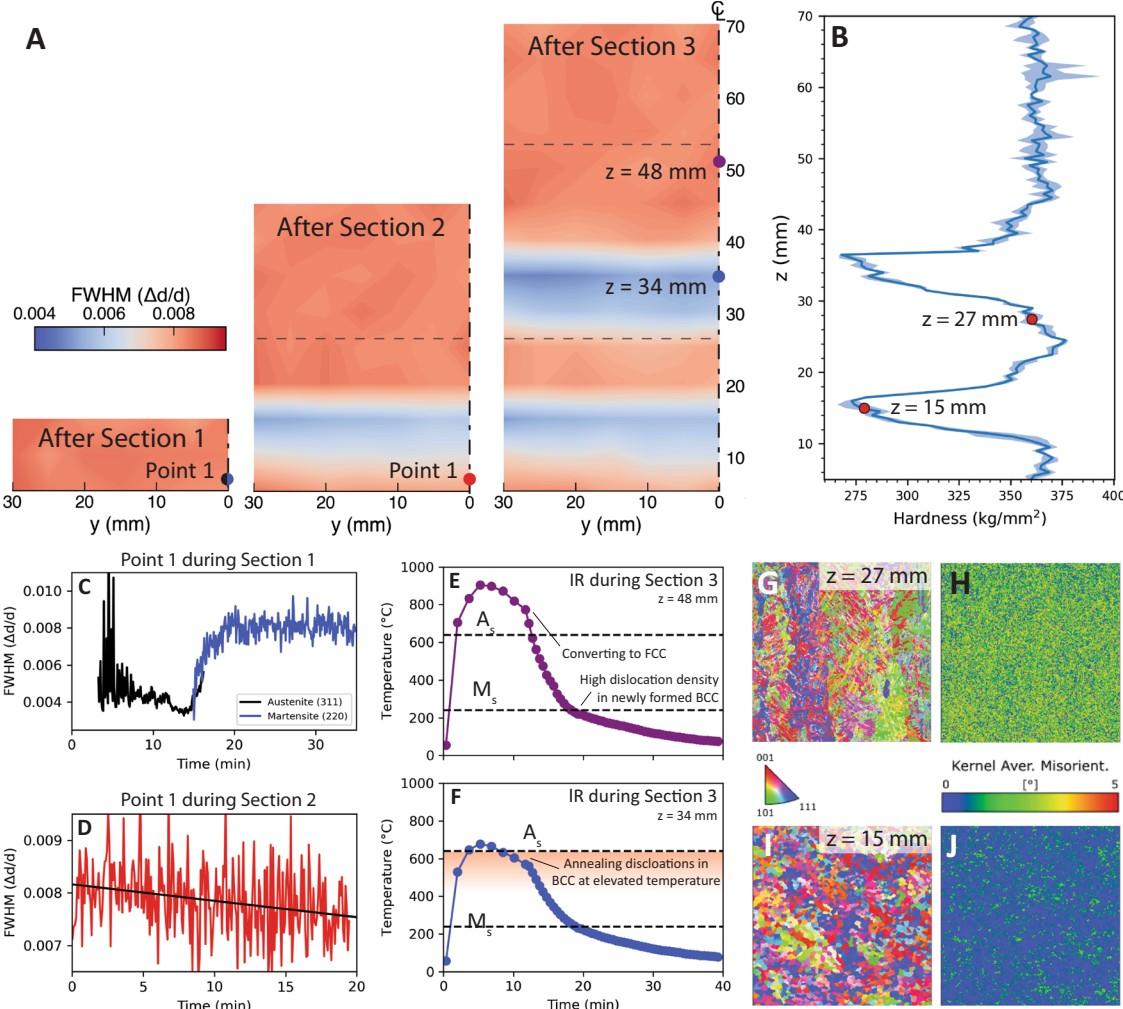

**Fig. 5 | Processing effects on plastic strain, microstructure, and hardness.**
**A** Distributions of the BCC full-width half-maximum (FWHM) through the sample measured at room temperature after building each sections which correlates with **B** high and low regions of microhardness measured along the sample centerline. **C** The time-resolved FWHM at point 1 increases during the transformation from FCC to BCC on cooling following the deposition of section 1 and **D** decreases when reheated during the deposition of section 2. **E** IR data at a height of 48 mm during building of section 3 shows a location with final cooling going through the FCC to BCC transformation at $M_s$ which correlates to a region of high FWHM and **F** a region at 34 mm, where the final thermal cycle heats to just over $A_s$, causing annealing and lower final FHWM. **G** The BCC grain structure for regions with high FWHM indicates lath martensite with **H** high kernel average misorientation, indicating high dislocation density. **I** Areas with low FWHM have a recrystallized grain structure (scale is 200 μm) and **J** very low kernel average misorientation.

sections, but not to a high enough temperature to form an appreciable FCC fraction. Reheating of the BCC structure anneals dislocations and reduces the FWHM[43]. Consequently, this variation in dislocation content depends on thermal history, and therefore the processing conditions. Importantly, because these regions are associated with temperatures near the phase transition between FCC and BCC, there is an inherent coupling of the distribution of dislocation density and the residual elastic strain.

The competing phase transformations (i.e., FCC to BCC and the annealing of BCC) and their influence on dislocation density have two effects that may be confirmed in the final material state (see Supplementary Note 2 for characterization details). First, the microstructure was evaluated using electron backscatter diffraction, which reveals the grain structure (Fig. 5G, I) and orientation gradients (Fig. 5H, J, showing kernel average misorientation maps) that result from dislocation content. Areas with high FWHM exhibits a lath martensite grain structure with high kernel average misorientation, consistent with high dislocation content. Regions with low FWHM, by comparison, show a recrystallized grain structure with low kernel average misorientation. Second, the local dislocation content affects the yield strength of the

material through forest hardening, which may be estimated by mapping the hardness along the height of the sample (Fig. 5B). In these data, the regions with low FWHM correlate with areas of low hardness. These observations are consistent with annealing of the martensitic structure due to cyclic reheating during fabrication.

## Discussion

The experimental observations imply that below the border between each pair of sections, there exists an interface between material that was reheated to a high enough temperature to convert from BCC to FCC, and material that remained BCC during the deposition process of the latter section. A coupled thermomechanical phase transformation model (see Supplementary Note 3) was used to gain additional insight into locations that were not probed during the operando experiments. Figure 6 shows the simulated longitudinal elastic strain during deposition and cooling of section 2. During deposition and the initial stages of cooling, the CTE mismatch at the FCC/BCC interface produces tensile strains in the FCC and compressive strains in the BCC. However, as the FCC cools to $M_s$, the phase transformation from FCC to BCC opposes and eventually reverses this strain field. As shown in

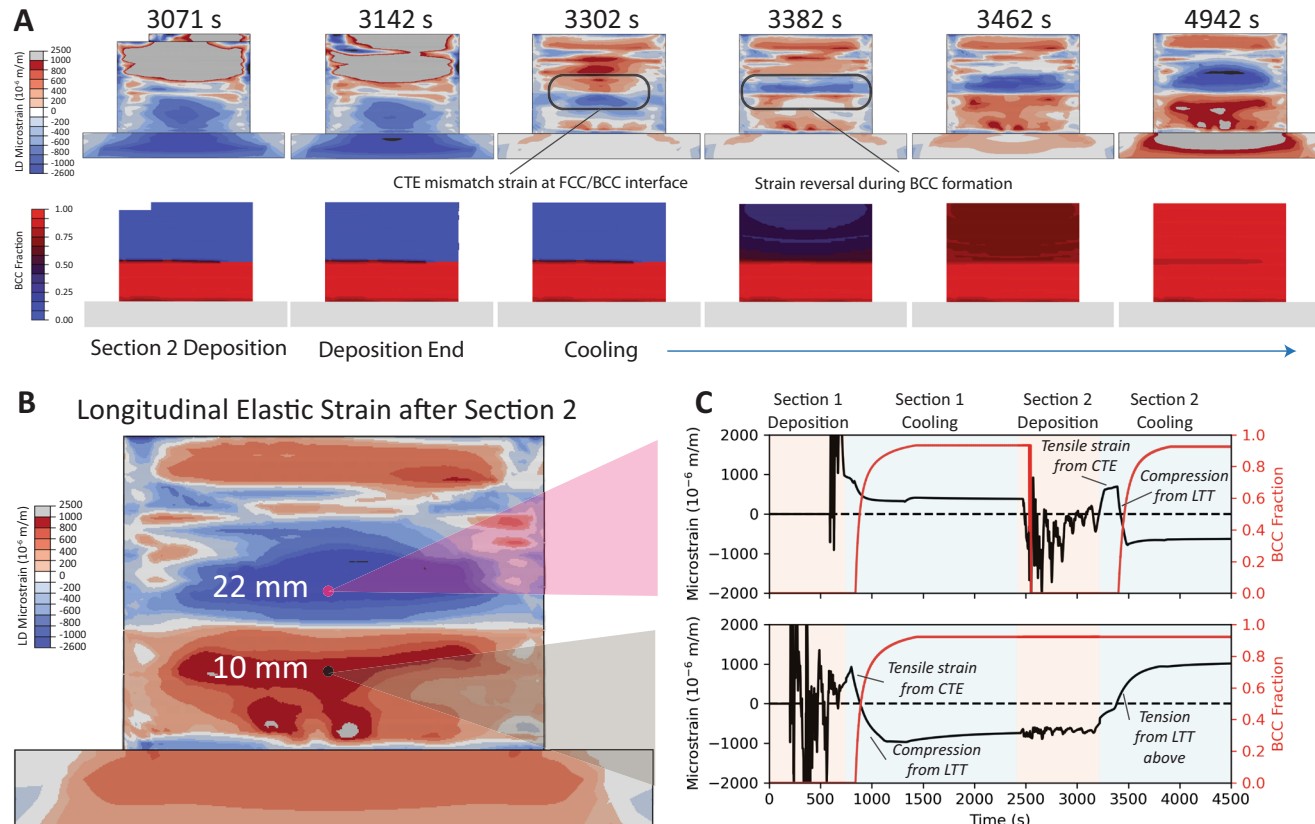

**Fig. 6 | Integrated thermomechanical phase-transformation model.**
**A** Simulated longitudinal elastic strain distributions and BCC phase fraction during and after deposition of section 2. **B** The final predicted longitudinal elastic strain distribution following cooling to room temperature and **C** selected time-series data for two points through sections 1 and 2 resulting in (top) compressive and (bottom) tensile residual elastic strains.

Fig. 6C, this sequence of events creates a distinct signature in the time-dependent strain evolution that depends on location relative to the FCC/BCC interface. On the FCC side, cooling following deposition first generates a spike in tensile strain due to thermal gradients and the CTE mismatch between phases, followed by the formation of compressive strains during the phase transformation. Notably, this trend is observed experimentally in the operando neutron data (Fig. 4) for locations cooling from an FCC state. Alternatively, locations below the FCC/BCC interface have an initially compressive longitudinal elastic strain upon cooling, which converts to a tensile strain following transformation of the FCC material above. Note that the transverse strains generally experience opposite trends in strain evolution but with a lower magnitude due to the Poisson effect.

The combination of operando neutron characterization data and validated model results shows an interrelationship between the process conditions, phase transformations, and lattice strain evolution. Importantly, high temporal and spatial variation in thermal and mechanical conditions plays a critical role in determining these effects, giving rise to a very different response for the LTT steel compared with idealized experiments[8]. Critically, reheating of previously deposited material from BCC back into the FCC phase field under the presence of a thermal gradient produces an interface between the two phases, the shape and motion of which dominates the elastic and plastic strain evolution (Fig. 7). As the final residual stress distribution is strongly dependent on gradients in the plastic strain distribution, elastic strains that drive deformation and thermal effects that modify the stored plastic strain play important roles. The observable effects towards residual stress development may therefore be described as follows. (1) Spatially varying thermal contraction in response to temperature gradients within the material drives elastic deformation, which would exist in the absence of the FCC/BCC interface. But strain at the interface develops as a function of time-dependent cooling caused by the CTE mismatch between phases, which further modifies and reinforces the effect of vertical thermal gradients, driving a strong but transient elastic strain field that manifests as tensile longitudinal elastic lattice strain in the FCC and compression in the BCC. (2) The volume expansion from FCC to BCC during cooling through the $M_s$ temperature causes a complete strain reversal at the FCC/BCC interface and affects the elastic strain distribution across the sample. (3) Plastic strain accumulation is caused by transient evolution of stresses which is dependent upon the temperature as it modifies both the local stress state and local mechanical properties. This effect is also influenced by the phase transformation as the formation of BCC from FCC necessarily accumulates large dislocation content. (4) Thermal annealing effects that modify the stored plastic strain, an effect that is particularly notable in the BCC just below $A_s$, where annealing is rapid, but transformation to FCC and back to BCC does not occur, which would accumulate new dislocation content. Notably, these effects are either directly dependent upon or strongly modified by the transient evolution of the FCC/BCC interface. This observation creates a direct causal link between the process conditions and geometry of the sample with the final residual stress state.

These observations have a significant impact on the performance of as-fabricated LTT steel. The inhomogeneous distribution of plastic strain directly affects the mechanical strength and strain hardening response of the material, as measured by the hardness distribution in Fig. 5J; and the elastic residual strain distribution modifies the effective stress experienced by the material at each location in response to an applied load. Based on the mechanisms for the formation of these distributions, it is clear that they are predominantly controlled by the time evolution of the FCC/BCC interface geometry. As a result, it is important to consider the parameters that might affect the evolution

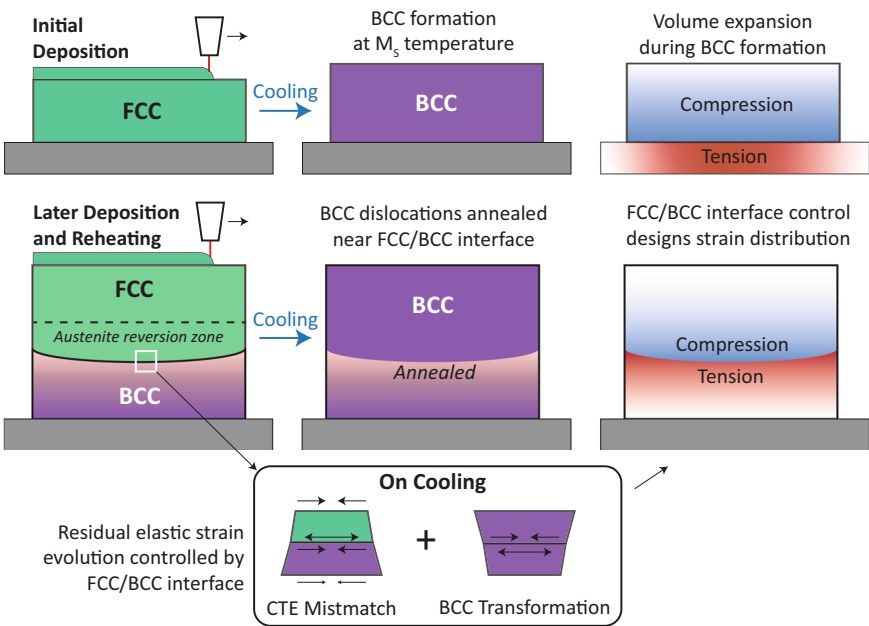

**Fig. 7 | Mechanism of residual strain evolution.** Residual stain evolution depends on heating, cooling, and reheating during deposition of multiple build sections. On initial deposition, expansion during the FCC to BCC formation creates a compressive strain in the deposit and tensile strain in the substrate. During reheating, an FCC/BCC interface forms, at which the differential CTE and the FCC to BCC transformation on cooling create a localized compression/tension strain pattern. In addition, reheating of BCC below the interface creates an annealing affect that softens the material.

of the interface. As shown here for a simple case, the heating, cooling, and reheating of the sample in response to the deposition process controls the interface location. However, this effect further depends on the deposition parameters and the sample boundary conditions, and therefore the geometry of the component being produced. As a result, there are both multiple controls available to be tuned (welding parameters, inter-pass and inter-layer timing, active cooling controls) and constraints (desired geometry, thermal properties of the deposited material). The results herein demonstrate that a balance of these controls and constraints may be used, with the guidance of computational modeling, to design and manipulate the final material state, including both properties and residual stress distributions.

## Methods
### Manufacturing and materials
A custom wire-arc AM (WAAM) system was designed and commissioned, consisting of a Lincoln Electric R450 metal inert gas (MIG) welder mounted as an end-effector on a Tormach ZA6 six-axis robotic arm. This system was used to fabricate planar walls in three sections (Fig. 1A) of seven layers each, using a bidirectional rotated scan path. Following the fabrication of each section, air was forced through a cross-drilled aluminum substrate to cool the sample to room temperature. Samples were fabricated from both mild steel and a custom LTT steel. See Supplementary information for deposition parameters and additional details.

### Neutron diffraction
Ex situ neutron residual strain maps of the samples fabricated by AM were measured at the high intensity diffractometer for residual stress analysis (HIDRA) at the High-Flux Isotope Reactor (HFIR) at Oak Ridge National Laboratory. Operando neutron diffraction was performed at the VULCAN beam line at Oak Ridge National Laboratory's Spallation Neutron Source. Neutron data were collected using a 5 mm cubic voxel in a time-resolved fashion during processing. During the intervals between sections, the build was cooled to room temperature and ND data were measured on a 5 mm regular grid to map lattice strain. Data was reduced to 1-dimensional intensity vs d-spacing for each detector

using a calibration for detector offset for each pixel. Reduction was performed at 1 or 5-s intervals for visualization and analysis. A strain-free reference lattice spacing for BCC at room temperature was measured from an annealed sample. A reliable reference could not be measured directly for FCC and instead, $d_0$ values were calibrated for each detector assuming equal temperatures for BCC and FCC within the diffracted voxel when both phases were present.

### Infrared data collection
The system was outfitted with a FLIR A700 long-wave infrared (IR) camera that collected data at 30 Hz with 640 × 480 resolution at 17 mm focal length resulting in 0.55 mm object pixel resolution. Data was filtered for a 10 μm wavelength using a FBXX000-500 bandpass filter.

### Materials characterization
The LTT wall was sectioned into two halves, each of which was prepared using standard metallographic procedures. Electron backscatter diffraction (EBSD) data was collected using a Zeiss Crossbeam 550 dual beam FIB/SEM with Oxford XMax EBSD detector. EBSD data was analyzed using MTEX. Hardness testing was performed using a LECO AMH 55 automatic hardness tester with a load of 1 kgf and dwell time of 15 s.

### Numerical modeling
The three-dimensional finite element model was developed in the Abaqus 2020 AM module[29,44,45]. The model was calibrated and validated directly against the neutron data using effective heat transfer coefficients for the aluminum substrate, including the effect of active cooling, to match the thermal profiles. The phase change was modeled using the K-M equation[46] and calibrated to be consistent with the ND observations (see supplementary material for details).

## Data availability
All data are available in the main text or the supplementary materials. Raw data will be made available upon request.

## Code availability
All analysis code will be made available upon request.

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

## Acknowledgements

The authors would like to thank Badri Narayanan from Lincoln Electric for providing the LTT feedstock material and Daniel Rogge from Tormach for support in integrating and commissioning the system. Tony Schmitz advised on the construction of the operando AM system, and Justin West provided preliminary mechanical design. OpenMIND Technologies and Roboris SRL also provided software support. This research was sponsored by the Laboratory Directed Research and Development Program of Oak Ridge National Laboratory, managed by UT-Battelle, LLC, for the US Department of Energy. A portion of this research used resources both at the Spallation Neutron Source and the High Flux Isotope Reactor, DOE Office of Science User Facilities operated by Oak Ridge National Laboratory.

## Author contributions

A.P.: Conceptualization, data analysis, writing, and editing; K.S.: Methodology, software, investigation, editing, supervision; C.S.: Conceptualization, data analysis, writing and editing; J.H.: Methodology, software, investigation, editing; G.M.: Software, data analysis, writing and editing; K.A.: Methodology, resources, editing; R.K.: Characterization, investigation, editing; T.F.: Methodology, investigation, editing; Y.L.: Methodology, software, supervision; D.Y.: Methodology, resources, investigation; C.L.: Software; J.V.: Software; SSB: Writing—review and editing.

## Competing interests

The authors declare no competing interests.
