## [Peer Review File · Nature Communications]

Operando neutron diffraction reveals mechanisms for controlled strain evolution in 3D printingREVIEWER COMMENTS

Reviewer #1 (Remarks to the Author):

This is a thought provoking paper showing the study of a WAAM build with in-situ neutron diffraction at the VULCAN instrument. While I've known several groups who talked about doing this for various AM techniques for the bragging rights of putting an AM machine in a beamline, this paper is the first I've seen that has done it to actually derive meaningful scientific conclusions. It will have considerable relevance to other groups attempting this for the study of similar processes, especially as in-situ neutron diffraction becomes an increasingly common tool for such studies.

Overall the method seems sound, and some initial concerns that I had about the method don't seem to invalidate the final conclusions. However, it took some effort reading the paper for me to convince myself of that fact. Thought should be given to the order in which ideas are first presented, and making sure they are presented adequately at that point. Several times, I find myself asking questions about the process that are only answered in passing later in the manuscript, or read descriptions that are simplified to the point of incorrectness until a full discussion of those topics takes place. Normally you'd try to avoid creating suspense or anticipation in scientific writing, but that might be a lesser evil compared to creating a misleading impression of something that is only corrected later in the paper.

There are two specific themes in this paper where that fault is a particular concern:

Causes of Residual Stress

One particular example here is in the specific ways that residual stress is created or modified by the process of the WAAM build. You say 'gradients in accumulated plastic strain' (P2L67) are the cause of residual stress. I can live with this simplification in the abstract, and you're safe in saying higher plastic strain gradients typically create higher residual stresses, but a gradient alone is not sufficient to cause residual stress; the induced plastic strain field must be incompatible, i.e. it creates a stress field such that no elastic displacement field could be superposed to reduce the strain energy. Later in the paper (P9L253) you talk about how the phase transformation affects the stress, but the discussion does not appear to consider that even a single phase material would have a very high stress field (probably higher albeit simpler) and the phase transformation is merely modifying this. The difference in CTE (P11L348) is not automatically a source of stress, would you not also see a similar effect in a material of uniform CTE just by virtue of the different temperatures present? It's not really a case of CTE mismatch like you would see in a bimetallic strip, since the interface is always at the temperature dictated by the curve in Fig 1D, at least in this mental model of the process. It might be helpful to have a specific point in the paper where you talk about the different (possibly competing or counteracting) mechanisms that affect both the transient (during cooling) stress and the final (residual) stress, which to my mind are (1) ordinary thermal contraction (2) plastic strain resulting from the transient stresses, an effect that is itself temperature dependent, (3) volume change on the FCC-BCC transition (is it isotropic?) (4) mismatch of the FCC and BCC expansion coefficients, and (5) thermal annealing of stress, which may be a special case

of (2) or indistinguishable from it. The discussion of (3) and (4) without reference to the others seems to miss the bigger picture in some places.

Use of Detector 3

The VULCAN detector 3 is at 150° 2θ , giving a measurement diffraction vector that is 60° away from the plane of the sample. Effectively you're assuming that there is zero stress normal to the surface (no harm in saying this explicitly – it's not enough to assume the principal direction is aligned with the sample axes), since under this condition I think you have zero strain at an angle of $\arctan(1/\sqrt{\text{Poisson's Ratio}})$ to the in-plane. For a Poisson's ratio of 0.3 this gives 61.3° . I guess to be most accurate you could chop down the solid angle of this detector to the angle that exactly matches your Poisson's ratio, but I don't think that improves accuracy significantly. Note that you still have the issue of vertical stress in the build direction possibly existing (I don't think you mention this in the paper at all), which would cause an equal effect via its Poisson strain on all the d-spacings measured by the three detectors. The longitudinal strain is therefore still correct, or more precisely becomes the value that the longitudinal strain would be in the absence of Poisson strain from any vertical stress, which is probably what you're really after anyway. It does become a potential issue when you use the lattice parameter to infer temperature, but the effect is probably small, and you explicitly quantify the possible error induced from this which is good.

My complaint is that I had to think about these arguments myself to decide it was all ultimately valid, and I'm still left wondering what the relevance of the stress-free d_0 mentioned in the supplementary material is (which are normally made by EDMing a comb sample to relieve stress in the teeth of the comb).

Other issues, wording and typos

The arrow -> below indicates a suggested change.

P1L18 "also offer": The complex conditions themselves do not offer this, the ability to control them does.

P1L38 apply -> induce ['apply' suggests external loads applied to a surface, i.e. in how 'applied stress' is in contrast to 'residual stress']

P2L45 'compressive strains' -> 'compressive stresses', since the strain is constrained to be zero in this example. Could say 'compressive elastic strains' if you want to stick with the word strain.

P2L63 Again, only say 'applied' here if you're talking about active cooling that is varied during the build process (since you talk about 'cycles'). I grant you may be, but I believe that's less common compared to e.g. fixed cooling of the base plate.

P2L64 characterization -> characterize

P2L65 'process' -> 'the process', or 'processing'.

P2Eq1 Should you include a phase transformation strain explicitly here?

P4L116 Use subscripted S in A_s for austenite start temperature, especially as otherwise it just reads as the word 'As'. (it may already be subscripted, sorry it's not obvious in the font used on my printout)

P4L124 5mm voxels -> $5 \times 5 \times 5 \text{ mm}^3$ voxels or 5mm cubic voxels. [you later reveal these are cubes]

P4L129-L135 Worth mentioning, but could remove this paragraph if needed to make space – most people don't worry about this effect when measuring steels!

P5Fig2 Clearly distinguish between labels of what data is being plotted, and what is being done at the time that data was measured. E.g. change 'Section X' heading to 'Section X build', and 'Point Y' label to 'Diffraction from Point Y'. What are Operando Build 1 and 2? Is the time axis relative to the start of building each Section?

P6L195 'upon activation of active cooling' Was the cooling not active all the time? I'm wondering if it's worth starting with a master time plot showing what was done at each point during the whole experiment.

P7Fig3 Are the labels B-D and E-G erroneously swapped in the caption here? Also, what does sigma refer to? If it's standard deviation of the measured temperature, is that deviation within some spatial window, or deviation over the time duration of some measurement interval?

P7L215 and Fig4. It's still unclear what was done here. Is this a measurement of the whole sample after the in-situ build, or was each section measured separately after it was created and cooled to room temperature? '...measured at room temperature following cooling of each of the three build sections'. I'm assuming it's not the second option, or you would have done Rietveld (being on VULCAN) instead of measuring just the 211 peak (implying you took the sample away to measure on a monochromatic source). Also in the figure 4 and B, why are the sections numbered top to bottom, and the section maps the different sizes that they are? Does the layout of this map no longer correspond to the layout of these areas in the sample itself?

P7Eqn3 and 4: What are you using for the d_0 term here? Isn't it just d_{03} as per your explanation in the paragraph above? In practice it makes little difference which kind of d measurement you use in the denominator for this equation, but if the same term is used twice then it should be called the same thing.

P9L271 "returned to a compressive strain": How can it return to a compressive strain when it was already at a compressive strain as you mention in L269? "longitudinal strain became compressive"

P10Fig5: Caption has no mention of colour map E. Is this strain or peak width? The term $\Delta d/d$ can refer to either depending on context. Also graphs A, C, D seem to be for different locations, but B is the same location as A so doesn't logically sit in the sequence. Why is there even an arrow to a point for graph B? Not only was it not measured there (according to the caption), it also wasn't measured just while that point was being built, since the time axis of the graph implies it corresponds to the build duration of the entirety of Section 2. Again, why the inversion of the order of the maps and graphs for Sections 1,2,3, is this a style issue for Nature journals and the order that subfigures are referred to?

P16 reference 48: Soothing -> Smoothing

In the Supplementary material

P1L3 title not quite the same as for the main article

P2L42 Use lower case k for kg.

P2L48 is Lincoln Electric one word?

P2L48 add 'with' after 'provided' if 'was' is the main verb of this sentence.

P3L59 of neutron -> for neutron [if I interpret this correctly]

P3 table S2 This seems to be the first mention of WAAMing mild steel! Is there more than one sample? Or are they just typical values provided for comparison?

P4L89 spacing -> spacings

P4L95 In general this is a bad way to get stress-free references since the thermal history itself changes d0! E.g. interstitial carbon vs carbides. The cooling history from this temperature would also be important, especially for an LTT steel.

P4L108 Use correct capitalised full name of GSAS-II, or can probably just say 'GSAS-II'

P5FigS3 How does temperature vs time for these same points look? Is the difference completely explained by the slower cooling for the higher points?

P5L126 GSAS-II name again

P5L145 You say CTE is constant for all temperatures – what range are you considering? Quadratic relation sometimes used here

P6L162 of section -> of that section [or whatever is meant]

P6L171 spacing -> spacings

P6L180 for sections -> for the building of sections

P8FigS5 For this type of plot consider being more explicit in labels e.g. 'Section 2' -> 'During Section 2 building' and Temperature -> temperature at point 1.

P13FigS10 Which colour line is which? Also could the IR pyrometry data be shown on these same graphs?

P14L316 on -> in

P14L335 austenite contracts -> austenite expands then contracts [you're talking about temperature movements in both directions, yes?] Or, change 'this period' to 'the cooling period'

Reviewer #2 (Remarks to the Author):

The paper describes in detail an first time in operando study with neutrons, connection between strain evolution and phase transformation. The work is definitely original, no one in the neutron strain community has attempted an experiment with the same level of complexity.

My recommendation is to publish the paper after a few issues are addressed and discussed.

A detailed review can be found in the attached report in PDF format.

Reviewer #3 (Remarks to the Author):

This paper deal with operando observation of mechanical strain, temperature and phase transformation behavior in LTT steel during additive manufacturing process using pulsed neutron diffraction. This is not only a highly novel attempt, but also provides valuable knowledge for the development of additive

manufacturing of metals and alloys, which is a relatively new material processing method. By combining multiple methods such as residual stress analysis using neutron diffraction, temperature distribution measurement using an infrared thermometer, and simulation, the behavior of materials is evaluated in more detail. This research will be of interest to many in the mechanical engineering, materials science and quantum beam communities. I think however that there are a few improvements that should be made before publication.

Comments are as follows:

- 1) Only strain in the LD and TD is evaluated. Are there no strains or residual stresses in the building direction? If they do not need to be considered, please indicate the reason.
- 2) It seems not enough clear to me why LTT steel is "representative": is it because the difference between Ms and As is so large that it is easy to evaluate phase transformation? How important is LTT as a practical additive manufacturing material? Are there any examples of practical applications? Please indicate more clearly why LTT steel is selected as the material for the evaluation in this study.
- 3) What criteria were used to divide the sections, and to determine point locations?
- 4) Please show a typical one-dimensional diffraction pattern (Intensity vs. d or TOF) and fitting results for the supplement.
- 5) What is the melting point of LTT steel and to what temperature does it increase during AM? Are those temperatures consistent with the results in Fig. 3?
- 6) The red text in Fig. 1 and Fig. 2 is too small to read.
- 7) It is unclear why the contribution of elastic strain to the strain obtained by the Q3 detector is small; is it because the tensile/compressive states are reversed for LD and TD?
- 8) Even without using the Q3 detector data, I think it is possible to estimate the thermal strain on Q1 and Q2 using the temperature measured by IR and CTE. What are the advantages of using Q3 data?
- 9) In Fig. 3, the temperatures determined from the peak shift and measured by IR are almost identical below 500°C. Is it correct that these temperatures coincide, since the neutron observes the inside of the sample and the IR observes the sample surface?
- 10) On page 8, line 237, shouldn't "in Figures 4B and C" be "4C and D"?
- 11) (E) is missing from the caption of Figure 5.
- 12) It states that the heat treatment to obtain d0 was performed at 800°C for 8 hours. Are there phase stresses generated by the phase transformation during cooling?

REVIEWER COMMENTS

Reviewer #1 (Remarks to the Author):

Comment: This is a thought provoking paper showing the study of a WAAM build with in-situ neutron diffraction at the VULCAN instrument. While I've known several groups who talked about doing this for various AM techniques for the bragging rights of putting an AM machine in a beamline, this paper is the first I've seen that has done it to actually derive meaningful scientific conclusions. It will have considerable relevance to other groups attempting this for the study of similar processes, especially as in-situ neutron diffraction becomes an increasingly common tool for such studies.

Overall the method seems sound, and some initial concerns that I had about the method don't seem to invalidate the final conclusions. However, it took some effort reading the paper for me to convince myself of that fact. Thought should be given to the order in which ideas are first presented, and making sure they are presented adequately at that point. Several times, I find myself asking questions about the process that are only answered in passing later in the manuscript, or read descriptions that are simplified to the point of incorrectness until a full discussion of those topics takes place. Normally you'd try to avoid creating suspense or anticipation in scientific writing, but that might be a lesser evil compared to creating a misleading impression of something that is only corrected later in the paper.

Response: Thank you for the detailed and informative review. We greatly appreciate the effort put forth to evaluate the manuscript and recommend changes that, we believe, have greatly improved the overall quality and clarity. On the point of organization, we have significantly rewritten the early portions of the manuscript where the experimental setup is introduced. We now include the relevant assumptions and approach to analysis up front. Previously we had avoided incorporating this more conventional approach to a methods section (methods are listed at the end of the manuscript in the Nature Communications format), but given that the measurement technique and analysis is a key component of the novelty of this work, we agree that this approach is more suitable and increases clarity.

Comment: Causes of Residual Stress. On particular example here is in the specific ways that residual stress is created or modified by the process of the WAAM build. You say 'gradients in accumulated plastic strain' (P2L67) are the cause of residual stress. I can live with this simplification in the abstract, and you're safe in saying higher plastic strain gradients typically create higher residual stresses, but a gradient alone is not sufficient to cause residual stress; the induced plastic strain field must be incompatible, i.e. it creates a stress field such that no elastic displacement field could be superposed to reduce the strain energy. Later in the paper (P9L253) you talk about how the phase transformation affects the stress, but the discussion does not appear to consider that even a single phase material would have a very high stress field (probably higher albeit simpler) and the phase transformation is merely modifying this. The difference in CTE (P11L348) is not automatically a source of stress, would you not also see a similar effect in a material of uniform CTE just by virtue of the different temperatures present? It's not really a case of CTE mismatch like you would see in a bimetallic strip, since the interface is always at the temperature dictated by the curve in Fig 1D, at least in this mental model of the process. It might be helpful to have a specific point in the paper where you talk about the different (possibly competing or counteracting) mechanisms that affect both the transient (during cooling) stress and the final (residual) stress, which to my mind are (1) ordinary thermal contraction (2) plastic strain resulting from the transient stresses, an effect that is itself temperature dependent, (3) volume change on the FCC-BCC transition (is it isotropic?) (4) mismatch of the FCC and BCC expansion coefficients, and (5)

thermal annealing of stress, which may be a special case of (2) or indistinguishable from it. The discussion of (3) and (4) without reference to the others seems to miss the bigger picture in some places.

Response: Thank you for this comment. It has helped to clarify our assessment of residual stress and the role played by the quantities measured in the operando neutron diffraction experiments. First, in the abstract and introduction, we have expanded the description of the relationship between final residual stress distributions, and the elastic and plastic strain evolution during processing. As you point out, residual stress can of course develop in the absence of the FCC/BCC phase transformations that are observed in the specific case here. The key role of the transformation in this case is in how it *modifies* these strain components (hopefully you do not mind that we have borrowed this language in the manuscript). Specifically, there is a component of strain evolution associated with spatial thermal gradients, which lead to variations in strain regardless of the FCC/BCC interface. However, the presence of differential CTE at the interface adds a transient time dependent effect as a function cooling rate that would not be present for a single phase material (assuming zero thermal gradient). The Discussion has been significantly rewritten to contextualize the phase transformation effects within the broader scope of residual stress evolution.

On the question of whether the volume change during the transformation from FCC to BCC was isotropic, we did evaluate the neutron data and final microstructure to assess whether a strong texture or variant selection was present but did not find any strong indicators. The microstructure characterization is shown in the supplementary materials.

Comment: Use of Detector 3. The VULCAN detector 3 is at 150° 2θ , giving a measurement diffraction vector that is 60° away from the plane of the sample. Effectively you're assuming that there is zero stress normal to the surface (no harm in saying this explicitly – it's not enough to assume the principal direction is aligned with the sample axes), since under this condition I think you have zero strain at an angle of $\arctan(1/\sqrt{\text{Poisson's Ratio}})$ to the in-plane. For a Poisson's ratio of 0.3 this gives 61.3° . I guess to be most accurate you could chop down the solid angle of this detector to the angle that exactly matches your Poisson's ratio, but I don't think that improves accuracy significantly. Note that you still have the issue of vertical stress in the build direction possibly existing (I don't think you mention this in the paper at all), which would cause an equal effect via its Poisson strain on all the d-spacings measured by the three detectors. The longitudinal strain is therefore still correct, or more precisely becomes the value that the longitudinal strain would be in the absence of Poisson strain from any vertical stress, which is probably what you're really after anyway. It does become a potential issue when you use the lattice parameter to infer temperature, but the effect is probably small, and you explicitly quantify the possible error induced from this which is good. My complaint is that I had to think about these arguments myself to decide it was all ultimately valid, and I'm still left wondering what the relevance of the stress-free d_0 mentioned in the supplementary material is (which are normally made by EDMing a comb sample to relieve stress in the teeth of the comb).

Response: We have significantly modified the section which describes the experimental setup to explain the assumptions in the analysis and our approach to quantifying the error caused by these assumptions. Previously, some of this description had been incorporated in the text in the place where the data was presented. In addition to moving this description to a more logical location within the manuscript, it has been edited for clarity and expanded. As for the effect of vertical strain, we assume that the Poisson

effect is approximately isotropic for the in-plane detects. Therefore, taking the difference between detectors 1 or 2 and detector 3 cancels out this effect. We believe this is reasonable given the relatively low strength of the texture in the as-fabricated grain structure. This discussion has been added to the main text, with additional information in supplementary, including a brief analysis of the magnitude of the vertical strains based on the computational model.

Comment: P1L18 “also offer”: The complex conditions themselves do not offer this, the ability to control them does.

Response: Corrected.

Comment: P1L38 apply -> induce [‘apply’ suggests external loads applied to a surface, i.e. in how ‘applied stress’ is in contrast to ‘residual stress’]

Response: Corrected.

Comment: P2L45 ‘compressive strains’ -> ‘compressive stresses’, since the strain is constrained to be zero in this example. Could say ‘compressive elastic strains’ if you want to stick with the word strain.

Response: Corrected.

P2L63 Again, only say ‘applied’ here if you’re talking about active cooling that is varied during the build process (since you talk about ‘cycles’). I grant you may be, but I believe that’s less common compared to e.g. fixed cooling of the base plate.

Response: Changed to “experienced”.

P2L64 characterization -> characterize

Response: Corrected.

P2L65 ‘process’ -> ‘the process’, or ‘processing’.

Response: Corrected.

Comment: P2Eq1 Should you include a phase transformation strain explicitly here?

Response: We did not analyze the phase transformation strain explicitly, but rather, through its implicit affects on elastic and plastic strains in the constituent phases. As it related to the neutron diffraction

analysis, we therefore prefer to keep the simplified equation. However, we have noted this simplification in the text.

Comment: P4L116 Use subscripted S in A_s for austenite start temperature, especially as otherwise it just reads as the word 'As'. (it may already be subscripted, sorry it's not obvious in the font used on my printout)

Response: Yes it is subscripted in the text, but uses a capital S which looks somewhat large. These occurrences have been changed to lowercase "s" to improve the distinction. The same change has been made for M_s .

Comment: P4L124 5mm voxels -> 5x5x5mm³ voxels or 5mm cubic voxels. [you later reveal these are cubes]

Response: Corrected.

Comment: P4L129-L135 Worth mentioning, but could remove this paragraph if needed to make space – most people don't worry about this effect when measuring steels!

Response: We have left this discussion in for now. If length becomes an issue according to the editorial team, we will move this paragraph to the supplementary materials.

Comment: P5Fig2 Clearly distinguish between labels of what data is being plotted, and what is being done at the time that data was measured. E.g. change 'Section X' heading to 'Section X build', and 'Point Y' label to 'Diffraction from Point Y'. What are Operando Build 1 and 2? Is the time axis relative to the start of building each Section? P6L195 'upon activation of active cooling' Was the cooling not active all the time? I'm wondering if it's worth starting with a master time plot showing what was done at each point during the whole experiment.

Response: Similar to comments on the setup and analysis of the diffraction data, this detail was not made clear enough at the point when the experimental setup was introduced. We have modified this section to make the use of active cooling between build sections more clear. Figure 1 was expanded to include a schematic description of the process sequence in conjunction with the neutron diffraction measurements and active cooling steps. Additionally, annotations in Figs. 2, 3, 4, and 5 have been modified or added to distinguish which data was collected "during" building of a given section, or "after" build of a given section. The layout of Figures 4 and 5 were also modified to improve the clarity of presenting the static lattice strain maps taken at room temperature, and the location of the time-resolved measurements. Hopefully, taken together, these changes make it clear where and when each piece of data is coming from.

Comment: P7Fig3 Are the labels B-D and E-G erroneously swapped in the caption here? Also, what does

sigma refer to? If it's standard deviation of the measured temperature, is that deviation within some spatial window, or deviation over the time duration of some measurement interval?

Response: Figure labels were erroneously swapped from an earlier version of the figure that had a different layout. Yes, sigma stands for standard deviation based on the collection of IR pixels within the 5 mm square surface patch and was calculated for each IR time step. The text has been modified for clarity.

Comment: P7L215 and Fig4. It's still unclear what was done here. Is this a measurement of the whole sample after the in-situ build, or was each section measured separately after it was created and cooled to room temperature? '...measured at room temperature following cooling of each of the three build sections'. I'm assuming it's not the second option, or you would have done Rietveld (being on VULCAN) instead of measuring just the 211 peak (implying you took the sample away to measure on a monochromatic source). Also in the figure 4 and B, why are the sections numbered top to bottom, and the section maps the different sizes that they are? Does the layout of this map no longer correspond to the layout of these areas in the sample itself?

Response: In Figure 4, the images in A and B were originally laid out top to bottom in chronological order. However, we have modified the layout to improve the clarity of presentation. The labeling and annotation of each sub-figure has also been expanded top help clarity when each data set was collection, for example, "After Section 1", or "Point 2 during Section 2". These annotations are also clarified by the addition of Figure 1E, showing the overall process and neutron data collection sequence. The lattice strain maps were indeed collected at room temperature (now stated explicitly in the figure annotations) and were calculated using a single-peak fit such that it is self-consistent with the calculation of the time-resolve operando data.

Comment: P7Eqn3 and 4: What are you using for the d_0 term here? Isn't it just d_{Q3} as per your explanation in the paragraph above? In practice it makes little difference which kind of d measurement you use in the denominator for this equation, but if the same term is used twice then it should be called the same thing.

Response: This question now pertains to the new Eq. 4. We have modified the notation to make clear which quantity is being used in each case. In the numerator, the reference is the transient d -spacing measured by detector 3, the notation for which is now: $d_{hkl}^{\widehat{Q}_3}(t)$. The denominator is a strain-free room temperature reference. For consistency, we have now modified this to be clear which detector measurement was used for this reference (e.g., $d_{hkl}^{0, \widehat{Q}_3}$). Of course, in ideal circumstances, the reference value should be identical across detectors and it should not make a difference which is used. However, in responding to another comment (see response to Reviewer 3), we discovered a minor calibration issue that affected only the time-resolved analysis in which used data from combinations of detectors (specifically detectors 2 and 3). In this case, the selection of the reference value does matter as they are slightly different. When simultaneously using multiple detectors, the difference may be simply accounted for by adding a calibration offset to the numerator as we now show in Eq. 4. In the end, this issue does not affect the trends in the data (which is our primary concern here) and simply makes the

data self-consistent between time-resolved and static measurements which use different combinations of detector data at SNS.

Comment: P9L271 “returned to a compressive strain”: How can it return to a compressive strain when it was already at a compressive strain as you mention in L269? “longitudinal strain became compressive”

Response: This was a typo. The longitudinal strain was compressive after the fabrication of Section 1. During Section 2, the longitudinal strain at this point first became tensile during the build, then returned to a compressive state upon cooling. The text has been modified appropriately.

Comment: P10Fig5: Caption has no mention of colour map E. Is this strain or peak width? The term $\Delta d/d$ can refer to either depending on context. Also graphs A, C, D seem to be for different locations, but B is the same location as A so doesn't logically sit in the sequence. Why is there even an arrow to a point for graph B? Not only was it not measured there (according to the caption), it also wasn't measured just while that point was being built, since the time axis of the graph implies it corresponds to the build duration of the entirety of Section 2. Again, why the inversion of the order of the maps and graphs for Sections 1,2,3, is this a style issue for Nature journals and the order that subfigures are referred to?

Response: The caption in Fig. 5 has been corrected, and the colormap has been modified for clarity with a label of “FWHM” to indicate it is a measure of the peak breadth. The layout has also been modified to make it more intuitive what each data plot means. Additional annotations have been added to indicate that static data was taken, for example “After Section 1”, and that each piece of time-resolved data was, for example, “Point 1 during Section 1”. The respective locations are indicated on the static room temperature maps for reference.

Comment: P16 reference 48: Soothing -> Smoothing

Response: Corrected

In the Supplementary material

Comment: P1L3 title not quite the same as for the main article

Response: Corrected.

Comment: P2L42 Use lower case k for kg.

Response: Corrected.

Comment: P2L48 is Lincoln Electric one word?

Response: No, it is not. This has been corrected.

Comment: P2L48 add 'with' after 'provided' if 'was' is the main verb of this sentence.

Response: The sentence was revised and shortened for clarity and concision.

Comment: P3L59 of neutron -> for neutron [if I interpret this correctly]

Response: Corrected.

Comment: P3 table S2 This seems to be the first mention of WAAMing mild steel! Is there more than one sample? Or are they just typical values provided for comparison?

Response: A very limited amount of data was provided for mild steel in Figure 1 as a baseline for comparison. The caption for Table S2 has been modified so that this reference is clear.

Comment: P4L89 spacing -> spacings

Response: Corrected.

Comment: P4L95 In general this is a bad way to get stress-free references since the thermal history itself changes d_0 ! E.g. interstitial carbon vs carbides. The cooling history from this temperature would also be important, especially for an LTT steel.

Response: A stress-free reference obtained by EDM stress relief is typically a preferred route. The scale of the gauge volume used in this work is large, $5 \times 5 \times 5 \text{ mm}^3$. As such, a d_0 sample with dimensions $>6 \times 6 \times 6 \text{ mm}^3$, or $\sim 8 \times 8 \times 8 \text{ mm}^3$ to offer safe margins, is necessary to ensure the GV is fully buried and the measured data are free from edge effect artifacts that would have introduced artificial peak shifts. However, this large sample would have posed risks that the machined sample was truly representative of the stress-free reference. Prior work by the authors has "cite Wei's work" identified that the LTT material used in this work had no segregation and that the thermal history did not affect A_s or M_s as evidenced by DSC and dilatometry data measured before and after multiple heating cycles. Therefore, a thermally "relieved" sample was preferable to a reference obtained by EDM. Additionally, the limitation of only capturing two principal strain directions precluded using a force balance to infer the stress-free reference.

Comment: P4L108 Use correct capitalised full name of GSAS-II, or can probably just say 'GSAS-II'

Response: The package GSAS-II was listed by mistake. Prior to submission, we noticed that the conversion of data from unnormalized gas data to normalized f_{xye} for use in GSAS-II was introducing artifacts. Therefore phase analysis was performed using the old implementation of GSAS. The supplementary text has been corrected to reflect this detail.

Comment: P5FigS3 How does temperature vs time for these same points look? Is the difference completely explained by the slower cooling for the higher points?

Response: Yes, this trend is visible in the temperature data plotted in Figure 3 in the main text. Reference to this fact has been added.

Comment: P5L126 GSAS-II name again

Response: Corrected, used "GSAS".

Comment: P5L145 You say CTE is constant for all temperatures – what range are you considering? Quadratic relation sometimes used here.

Response: The CTE was calculated based on the dilatometry data in Figure 1, and for the purpose of the error analysis, a linear approximation was reasonable. The FCC is valid between M_s and approximately 1000 °C, BCC between room temperature and A_s . These details have been added.

Comment: P6L162 of section -> of that section [or whatever is meant]

Response: Corrected.

Comment: P6L171 spacing -> spacings

Response: Corrected.

Comment: P6L180 for sections -> for the building of sections

Response: Corrected.

Comment: P8FigS5 For this type of plot consider being more explicit in labels e.g. 'Section 2' -> 'During Section 2 building' and Temperature -> temperature at point 1.

Response: The caption for Fig. S5 was modified. Annotations in the figures in the main text were also adjusted appropriately.

Comment: P13FigS10 Which colour line is which? Also could the IR pyrometry data be shown on these same graphs?

Response: A legend has been added indicated the source of the data sets. In this case, it does not make sense to add the IR data to the plot, because the FEM data is homogenized across a volume that corresponds directly to the diffracted neutron voxel, and, at least during deposition, high gradients from the surface to that volume could cause discrepancies that are not representative of the quality of agreement between IR and the simulation results. The reason for this homogenization is that it is necessary for evaluating the phase change model accurately against the volume averaged phase fractions extracted from Rietveld refinement of the time-resolved diffraction data (Fig. S12). The simulation data is therefore averaged to the same scale as the neutron data. The figure caption has been updated to make this nuance clear.

Comment: P14L316 on -> in

Response: Corrected.

Comment: P14L335 austenite contracts -> austenite expands then contracts [you're talking about temperature movements in both directions, yes?] Or, change 'this period' to 'the cooling period'

Response: Correct to "the cooling period".

Reviewer #2 (Remarks to the Author):

Comment: The paper describes a first time, very thorough in-operando neutron diffraction measurement during additive manufacturing (AM). AM is a process method which promises essentially a “first time right” approach in designing components with right properties through optimisation of process parameters. In this respect, this is also a timely study as interest in AM industry now moves to real time investigation, which are key to understand and eventually control process and material response – and here especially the evolution of process inherent residual stresses. The paper connects the strain evolution throughout the build process to the effects of phase transformation in the used LTT steel. In this case, the authors rightly point out that their findings might enable manufacturing processes to be optimized so that residual stress distributions might be linked and controlled to favourable locations in the components. A combination of challenging experiments, which were not really attempted before in the community, together with simulations is used to rationalise the mechanism of strain development in that steel. The paper is clearly new and original and an excellent example what neutron diffraction is capable of. It is well written, very clear and with enough evidence to support the main conclusions. Therefore, I believe the paper is worthy to be published in Nature Communications.

Response: Thank you for the encouraging overview remarks. Please see below for detailed responses to your comments.

Comment: 1) The whole analysis approach hinges on the fact that one of the measured lattice spacing – here along Q3 – is only affected by thermal expansion so that it can be used as kind of reference to separate thermal and mechanical strain contributions. In line 166/167 there is only a very short and somewhat unsatisfactory statement given for that. Agreed, the T behaviour calculated on the basis of that d-values seems to compare quite well to the IR measured data and might serve as a justification. However, I would expect a bit more discussion concerning this statement – i.e. like assuming a linear $\sin^2\psi$ behaviour from compressive/transversal to tensile/longitudinal the values at Q3 could well be very small.

Response: This discussion on this analysis approach has been expanded and also moved to a more logical point within the manuscript so that it is not lost within the presentation of data later on. The key point is the argument that detector 3 should see little in-plane strain caused by the Poisson effect. Because the VULCAN detector setup does not allow us to interrogate the vertical axis, we must assume this component to be zero for the sake of analysis, with potential for corresponding errors. These effects are accounted for in our error analysis utilizing the IR data, and the latter point has also been expanded upon in the supplementary materials.

Comment: 2) The gauge volume (GV) is necessarily quite large in order to achieve the time resolution. However, that comes with some costs, as the strain is averaged across that volume of the GV it compares unfavourably to the thickness of the wall of the sample component and at least in the transverse direction one could expect low stress values. In addition, due to the large size of the GV volume one would also expect some T gradients within, which most likely also increases the error bars in the T calibration and/or the derived strains.

Response: Yes, you are correct, that the relatively large gauge volume has a significant disadvantage of potentially containing large thermal gradients. Indeed, this effect can be seen in the analysis of the IR data (e.g., Fig. 3B) where the standard deviation of the temperature for the pixels overlaying the GV is relatively large during process. Thankfully, the gradients shallow out significantly during cooling, where the key aspect of lattice strain evolution takes place.

In our diffraction analysis, the potential effect of gradients within the GV are accounted for in two ways. First, mean error in the d-spacing will manifest as potential errors relative to the IR data, as the gradient would be present in the difference in temperature from the surface of the sample (where IR data is measured) to the GV. Therefore, the error analysis accounts for this source of error. However, if the gradient within the GV is large without affecting the mean d-spacing, this manifests as an increase in the peak breadth. This effect is accounted for in the analysis of the FWHM presented in Figure 5C. As you can see, the FWHM is high and fluctuates wildly during processing, but collapses once building is finished and cooling begins. These effects have been noted in the section describing the error estimation and added to the text preceding Fig. 5, respectively.

Comment: 3) Fig. 4: As the layers essentially are cooled down after around 30 min (see the results on the cooling curves in Fig. S2) should the strain values of the ex-situ and in-situ measurements not match? As it seems to be not the same, please comment on that, as one also sees only compressive behaviour in both strain directions at point 3 in Fig 4C and D.

Response: This comment turns out to be a fine but important point. The error was the consequence of a difference in d_0 between detectors. Normally, when strain is calculated for a single detector based on the d-spacing shift for a specific peak, it is correct to use:

$$\varepsilon = \frac{d - d_0}{d_0}$$

where d_0 is a strain-free room temperature reference. In ideal circumstances, the d_0 value is a constant for the material and should be measured identically by each detector. In performing the analysis here, we found that detector 2 had a slightly different d_0 reference than the other detectors. Even so, using a consistent reference value for the given detector gives reasonable results because as it is self-consistent with the measurements of the AM sample. However, in our case, we are calculating the time-resolved data by combining data from two detectors, e.g.:

$$\varepsilon(t) = \frac{d^{\widehat{Q}_2} - d^{\widehat{Q}_3}}{d_0}$$

Here, the slight difference in reference between the two detectors creates an artificial offset between the d-spacing in the numerator which might be attributable to issues with the reference sample (unlikely here, as the same sample was measured at HFIR without issue), or minor errors in calibration between detectors.

Thankfully, this issue is easily resolved. We assume for this purpose that the calibration offset measured in the strain-free reference sample may be uniformly applied across measurements by simply applying this offset to cases where data from two detectors is compared. Consequently, Equation 4 is now expressed as:

$$\varepsilon_{el}^{L,D} = \frac{d_{hkl}^{\widehat{Q}_1}(t) - d_{hkl}^{\widehat{Q}_3}(t) + \Delta d_{hkl}^0}{d_{hkl}^{0,\widehat{Q}_3}},$$

where Δd_{hkl}^0 is a calibration offset expressing the difference between detectors. You will see that this correction does not affect trends in any of the data and therefore has no consequence for any of the major conclusions of the work. Because the issue only arose for time-resolved lattice strain analysis using a combination of detectors 2 and 3, the major effect is that the time-resolved transverse data in Figure 4D is now self-consistent with the final room-temperature transverse strain distribution in Figure 4B. Data within the paper has been updated appropriately, and a discussion on this point has been added to the supplementary materials to explain the use of the calibration offset.

Minor points:

Comment: Line 179: Change “... Figure 3B though ...” to “... Figure 3B through ...”

Response: Corrected.

Comment: Line 237: Change text to “... in Figures 4C and D...”

Response: Corrected.

Comment: Line 243/244: What is meant with “... Peak shift measured at 5 mm after deposition ...”. Do you mean a 5 mm step size between measurement points or a corresponding measurement grid? Please clarify that.

Response: Changed to “... measured with a 5mm grid spacing...”

Comment: Figure caption for Fig. 5: Explanation to what is figure part E is missing in the caption

Response: Added text.

Comment: Line 316: Change “...(Fig. 5F) ...” to “...(Fig. 5J) ...”

Response: Corrected.

Reviewer #3 (Remarks to the Author):

Comment: This paper deal with operando observation of mechanical strain, temperature and phase transformation behavior in LTT steel during additive manufacturing process using pulsed neutron diffraction. This is not only a highly novel attempt, but also provides valuable knowledge for the development of additive manufacturing of metals and alloys, which is a relatively new material processing method. By combining multiple methods such as residual stress analysis using neutron diffraction, temperature distribution measurement using an infrared thermometer, and simulation, the behavior of materials is evaluated in more detail. This research will be of interest to many in the mechanical engineering, materials science and quantum beam communities. I think however that there are a few improvements that should be made before publication.

Response: We thank the reviewer for the positive response to this work, as well as valuable review comments. Please see below for detailed responses to each.

Comment: 1) Only strain in the LD and TD is evaluated. Are there no strains or residual stresses in the building direction? If they do not need to be considered, please indicate the reason.

Response: The detector configuration at VULCAN limits our analysis to in-plane effects. Therefore, for the purpose of the analysis, we assume the vertical strains are small. We have addressed this comment in two ways. First, the assumption is now stated explicitly in the text introducing the experimental setup and describing the neutron diffraction analysis procedure. Note that errors accrued by neglecting the vertical strain component are captured by our approach to error estimation. Second, we have used the computational model (validated against the ND data) to quantify the amount of vertical strain and to compare it to the other component, which is now included in the Supplementary Materials. In short, the vertical strain is small compared to the longitudinal component, but not always compared to the transverse component. However, it only affects the validity of the analysis through the Poisson effect. Assuming a Poisson ratio of $\nu = 0.3$, we determine that the trends observed in the neutron diffraction analysis as well as the errors assessed experimentally are reasonable. This discussion has been added to the supplementary materials.

Comment: 2) It seems not enough clear to me why LTT steel is "representative": is it because the difference between Ms and As is so large that it is easy to evaluate phase transformation? How important is LTT as a practical additive manufacturing material? Are there any examples of practical applications? Please indicate more clearly why LTT steel is selected as the material for the evaluation in this study.

Response: It is not so much that LTT steels are "representative", but rather than it is a class of material with a particular phase transformation that has a significant effect on residual stress. Previous research has demonstrated that this might be a useful approach for AM. The work here may help to accelerate adoption of these materials for the purpose. The same technique may of course be applied to other material as well, which are likely to have their own mechanisms for affecting the final residual lattice strain distribution that could be the same or different from the LTT example.

Comment: 3) What criteria were used to divide the sections, and to determine point locations?

Response: Sections were each a set number of layers. The point locations were necessarily selected a priori. The rationale for point location was to make sure to capture the transformation from FCC to BCC upon cooling, which was most likely to occur very near the section boundaries. Unfortunately, this does mean that we missed measurements at some locations that might have been otherwise informative. The validated numerical model helps to fill in these gaps. The text in the manuscript has been modified to clarify these points.

Comment: 4) Please show a typical one-dimensional diffraction pattern (Intensity vs. d or TOF) and fitting results for the supplement.

Response: Added to the supplementary materials.

Comment: 5) What is the melting point of LTT steel and to what temperature does it increase during AM? Are those temperatures consistent with the results in Fig. 3?

Response: The melting temperature for the LTT steel is above the maximum shown for the IR data in Figure 3. The reason the temperature readings in Figure 3 do not range up to that temperature is because (a) the IR readings were taken with a specific bandpass filter during a filter wheel sweep performed several seconds after each layer was finished printing, and (b) the neutron diffraction data integrates over a 5mm voxel and can only diffract from solidified material, which will average out peak temperatures as well as show large variance caused by the locally high thermal gradients. Additionally, poor performance of the IR reading at temperatures near melting is expected, due to the large difference in emissivity of a liquid metal and oxidized surface, as well as range limits on the calibration performed. For these reasons, we do not rely on the IR or neutron diffraction data analysis at these high temperature for our interpretation of the experiments. Clarification in the caption, text, and supplementary to this effect was added.

Comment: 6) The red text in Fig. 1 and Fig. 2 is too small to read.

Response: Corrected.

Comment: 7) It is unclear why the contribution of elastic strain to the strain obtained by the Q3 detector is small; is it because the tensile/compressive states are reversed for LD and TD?

Response: We have significantly expanded the description of the neutron data analysis and moved it to a more logical position within the manuscript at the point where the experimental setup is introduced.

In short, the contribution of the Poisson effect from the longitudinal and transverse strains on the in-plane lattice strain measured by detector 3 is indeed expected to be small. This, however, does not account for contributions from vertical strain components, which cannot be directly measured with the detector configuration at VULCAN. We have added a section to the supplementary materials evaluating the magnitude of this effect based on the simulation results. Please note as well that the error analysis is designed to capture these effects.

Comment: 8) Even without using the Q3 detector data, I think it is possible to estimate the thermal strain on Q1 and Q2 using the temperature measured by IR and CTE. What are the advantages of using Q3 data?

Response: The advantage is that it is the same gauge volume at the same position and time. IR is at a different resolution (spatial and temporal). Data processing is therefore more straightforward and accurate to use the Q3 data as a temperature reference. Importantly, it also accounts for the Poisson effect that might be introduced from vertical strains. The IR then effectively supplements this data for error estimation. These differences between IR and the Q3 gauge volume temperature also mean that the error plotted for the transient elastic strain should be considered conservative, that is, it is likely an overestimate of the error. This rationale has been added to the supplementary text.

Comment: 9) In Fig. 3, the temperatures determined from the peak shift and measured by IR are almost identical below 500°C. Is it correct that these temperatures coincide, since the neutron observes the inside of the sample and the IR observes the sample surface?

Response: It suggests that the thermal gradients from the surface (IR measurement) and the GV are small, which is a good result, as it also means we may treat the GV as volume averaged for our measurements. This point has now been highlighted in the text describing the results shown in Figure 3.

Comment: 10) On page 8, line 237, shouldn't "in Figures 4B and C" be "4C and D"?

Response: Corrected.

Comment: 11) (E) is missing from the caption of Figure 5.

Response: Corrected.

Comment: 12) It states that the heat treatment to obtain d0 was performed at 800°C for 8 hours. Are there phase stresses generated by the phase transformation during cooling?

Response: Stresses at the interfaces between FCC and BCC would arise during the transformation. BCC nucleates with 24 possible variants within the FCC matrix. In the absence of an applied global strain which can preferentially select specific variants, nucleation results in an isotropic stress that would

overall cancel, resulting in a bulk material that does not have a residual stress. However, the resulting transformation strain induces significant dislocation content that increases the observed peak breadth in the measured diffraction data. But this effect does not impact the measurement of the reference lattice spacing. We note in the supplementary that this is an assumption in the approach to measuring the reference values, although, we believe, a good one, as even in the case of the AM sample itself, no strong variant select was identified.

Yes, a transformation strain may be caused by the conversion of FCC to BCC. However, these stresses are only visible in the diffraction data via their effect on either the elastic (i.e., d-spacing shift) or plastic (i.e., peak breadth) strain contributions. For this reason, we have chosen to maintain the simplified description of the contributions to the strain, but note that this is a simplification in the text that implicitly includes these transformation effects.

REVIEWERS' COMMENTS

Reviewer #1 (Remarks to the Author):

The revised paper addresses the concerns I had before and now reads much better. I especially appreciate the yellow highlights of the edited portions, and the inclusion of figure 1E. Most of the points I've noticed are just typos or grammar suggestions, but I have slight concerns about terminology (especially stress versus strain) in some places that risk misleading the reader:

P2L61 "controlling residual stress distribution" -> "controlling the residual stress distribution" or "controlling residual stress distributions"

P2L65: Similarly, insert "The" at start of sentence "residual stress distribution..."

P3L96: "along the height" -> "over the height" [not wrong, just sounds more natural to me!]

P4Fig1E: You could add your HFIR measurements at end of this, but no need if you think that would complicate things.

P5L148: "strains" -> "stresses", since Poisson effects are strains that arise from stresses in another direction.

P5L149... The reason the strain in detector 3 can be assumed negligible is not really that the Poisson strains are small, but that the overall strain is small by virtue of this direction lying between the stress direction (longitudinal) and a Poisson direction (transverse). Maybe change:

"This assumption is true only if Poisson effects from the longitudinal, transverse, and vertical strains are sufficiently small. In plane, the strain-free direction of the Poisson effects may be calculated as [Eq3] where $\bar{\nu}$ is Poisson's ratio. For $\nu = 0.3$, the zero-strain orientation is $T = 61.3^\circ$, which is very close to the 60° position of Q3 with respect to Q1."

...to something like...

"This assumption takes advantage of the stress in the transverse direction being negligible, and the strain in that direction being the Poisson strain from stress in the longitudinal direction. As this Poisson strain is opposite in sign to the longitudinal, there will be a direction θ from the longitudinal direction towards the transverse direction where the strain is zero, where [Eq3]..."

P5L150: "Vertical strains" -> "Vertical stresses". [Vertical strains are neglected simply by virtue of the fact that the detectors aren't oriented to measure strain in that direction. You're referring here to longitudinal and transverse strains caused by vertical stresses. See also point for P6L165 below].

P5L159: I didn't realise you were only finding strains from one peak in your Vulcan data. Why didn't you use Rietveld? Also if you're only ever using one hkl peak to find strain, you could omit the hkl subscript from your d terms for clarity, perhaps instead then making the 0 a subscript instead of superscript in the d0 terms.

P6L165 "Normal elastic strain" -> "Elastic strain" or even "Any elastic strain... would manifest as...". Diffraction always measures normal strain (as opposed to shear strain) and the word "normal" risks confusion with "normal to the sample surface".

After this add something like "Stress in the vertical (build) direction could be a minor cause of this. This stress is otherwise not considered in this work, as its effect is simply to superpose a small Poisson strain that is measured equally by the three detectors."

P6L168 "in detector 3" -> "made with data from detector 3"

P6L168 comma after "IR data"

P6L185 "compoition" -> "composition"

P14L406: In this added paragraph, points 1 and 2 have a main verb ("drives" and "causes" respectively), but 3 and 4 have noun phrases (with no main verb) as their first sentences. Either is fine, but the choice should be consistent for all 4 points.

P16L480 "lod" -> "load"

P16L485 reference 48 is missing from the reference list.

Reviewer #2 (Remarks to the Author):

The authors have addressed all open issues in full. As stated in my previous review, the paper is an excellent example of an experimental work using neutrons which will be of great interest to other scientists and engineers working in the field of AM. It should now be published as is.

Reviewer #3 (Remarks to the Author):

The authors have satisfactorily responded to all of my comments for previous version. I think this manuscript would be acceptable if the following minor points were corrected.

-The notation such as "(311) reflection" is not correct. (hkl) indicates the crystal plane indices, and "()" is not necessary for the diffraction indices.

-I am not sure what "in-strain plane" means on page 16 of the supplement. Perhaps this could be "in-plane stress".

REVIEWERS' COMMENTS

Reviewer #1:

Comment: The revised paper addresses the concerns I had before and now reads much better. I especially appreciate the yellow highlights of the edited portions, and the inclusion of figure 1E. Most of the points I've noticed are just typos or grammar suggestions, but I have slight concerns about terminology (especially stress versus strain) in some places that risk misleading the reader.

Response: Thank you for the positive feedback. We have addressed detailed comments as described below.

Comment: P2L61 "controlling residual stress distribution" -> "controlling the residual stress distribution" or "controlling residual stress distributions"

Response: Changed to the first option.

Comment: P2L65: Similarly, insert "The" at start of sentence "residual stress distribution..."

Response: Corrected.

Comment: P3L96: "along the height" -> "over the height" [not wrong, just sounds more natural to me!]

Response: Corrected.

Comment: P4Fig1E: You could add your HFIR measurements at end of this, but no need if you think that would complicate things.

Response: Chose to retain the current version of the figure to keep things simple. The caption does state specifically that the HFIR data is along the centerline.

Comment: P5L148: "strains" -> "stresses", since Poisson effects are strains that arise from stresses in another direction.

Response: Corrected.

Comment: P5L149... The reason the strain in detector 3 can be assumed negligible is not really that the Poisson strains are small, but that the overall strain is small by virtue of this direction lying between the stress direction (longitudinal) and a Poisson direction (transverse). Maybe change:
"This assumption is true only if Poisson effects from the longitudinal, transverse, and vertical strains are sufficiently small. In plane, the strain-free direction of the Poisson effects may be calculated as [Eq3] where ν is Poisson's ratio. For $\nu = 0.3$, the zero-strain orientation is $T = 61.3^\circ$, which is very close to the 60° position of Q3 with respect to Q1."

...to something like...

"This assumption takes advantage of the stress in the transverse direction being negligible, and the strain in that direction being the Poisson strain from stress in the longitudinal direction. As this Poisson strain is opposite in sign to the longitudinal, there will be a direction theta from the longitudinal direction towards the transverse direction where the strain is zero, where [Eq3]..."

Response: We rephrased this paragraph for clarity similar to the suggestion here.

Comment: P5L150: "Vertical strains" -> "Vertical stresses". [Vertical strains are neglected simply by virtue of the fact that the detectors aren't oriented to measure strain in that direction. You're referring here to longitudinal and transverse strains caused by vertical stresses. See also point for P6L165 below].

Response: Corrected.

Comment: P5L159: I didn't realise you were only finding strains from one peak in your Vulcan data. Why didn't you use Rietveld? Also if you're only ever using one hkl peak to find strain, you could omit the hkl subscript from your d terms for clarity, perhaps instead then making the 0 a subscript instead of superscript in the d0 terms.

Response: Using single peak fitting simplified the analysis process and avoid using an anisotropic elastic strain model within a Rietveld analysis to account for strain accumulation during the deposition process. We have chosen to simplify the notation as suggested.

Comment: P6L165 "Normal elastic strain" -> "Elastic strain" or even "Any elastic strain... would manifest as...". Diffraction always measures normal strain (as opposed to shear strain) and the word "normal" risks confusion with "normal to the sample surface". After this add something like "Stress in the vertical (build) direction could be a minor cause of this. This stress is otherwise not considered in this work, as its effect is simply to superpose a small Poisson strain that is measured equally by the three detectors."

Response: Corrected.

Comment: P6L168 "in detector 3" -> "made with data from detector 3"

Response: Corrected.

Comment: P6L168 comma after “IR data”

Response: Corrected.

Comment: P6L185 “compoition” -> “composition”

Response: Corrected.

Comment: P14L406: In this added paragraph, points 1 and 2 have a main verb (“drives” and “causes” respectively), but 3 and 4 have noun phrases (with no main verb) as their first sentences. Either is fine, but the choice should be consistent for all 4 points.

Response:

Comment: P16L480 “lod” -> “load”

Response: Corrected.

Comment: P16L485 reference 48 is missing from the reference list.

Response: References updated.

Reviewer #2:

Comment: The authors have addressed all open issues in full. As stated in my previous review, the paper is an excellent example of an experimental work using neutrons which will be of great interest to other scientists and engineers working in the field of AM. It should now be published as is.

Response: Thank you very much for the positive feedback. We are proud of this work and excited to share it with the community.

Reviewer #3:

Comment: The authors have satisfactorily responded to all of my comments for previous version. I think this manuscript would be acceptable if the following minor points were corrected.

Response: Thank you for the comments.

Comment: The notation such as "(311) reflection" is not correct. (hkl) indicates the crystal plane indices, and "()" is not necessary for the diffraction indices.

Response: Notation was corrected throughout.

Comment: I am not sure what "in-strain plane" means on page 16 of the supplement. Perhaps this could be "in-plane stress".

Response: Corrected.